# SALVE: Sparse Autoencoder-Latent Vector Editing for Mechanistic Control of Neural Networks

## Abstract

Deep neural networks achieve impressive performance but remain difficult to interpret and control. We present **SALVE** (Sparse Autoencoder-Latent Vector Editing), a unified "discover, validate, and control" framework that bridges mechanistic interpretability and model editing. Using an $\ell_1$-regularized autoencoder, we learn a sparse, model-native feature basis without supervision. We validate these features with Grad-FAM, a feature-level saliency mapping method that visually grounds latent features in input data. Leveraging the autoencoder's structure, we perform precise and permanent weight-space interventions, enabling continuous modulation of both class-defining and cross-class features. We further derive a critical suppression threshold, $\alpha_{\mathrm{crit}}$, quantifying each class's reliance on its dominant feature, supporting fine-grained robustness diagnostics. Our approach is validated on both convolutional (ResNet-18) and transformer-based (ViT-B/16) models, demonstrating consistent, interpretable control over their behavior. This work contributes a principled methodology for turning feature discovery into actionable model edits, advancing the development of transparent and controllable AI systems.

## 1 Introduction

Understanding the internal mechanisms of deep neural networks remains a central challenge in machine learning. While these models achieve remarkable performance, their opacity hinders our ability to trust, debug, and control their decision-making processes, especially in high-stakes applications where reliability is non-negotiable. The field of Mechanistic interpretability aims to resolve these issues by reverse-engineering how networks compute, identifying internal structures that correspond to meaningful concepts and establishing their influence on outputs (4; 23; 42; 1; 32) However, the bridge between interpretation and intervention remains a critical frontier. While recent advances in model steering successfully use discovered features to guide temporary, inference-time adjustments, a path toward using these insights to perform durable, permanent edits to a model's weights is less established. This paper closes that gap by introducing **SALVE** (Sparse Autoencoder-Latent Vector Editing), a unified framework that transforms interpretability insights into direct, permanent model control. We build a bridge from unsupervised feature discovery to fine-grained, post-hoc weight-space editing. Our core contribution is a "discover, validate, and control" pipeline that uses a sparse autoencoder (SAE) to first learn a model's native feature representations and then leverages that same structure to perform precise, continuous interventions on the model's weights.

Our framework achieves this by first training an $\ell_1$-regularized autoencoder on a model's internal activations to discover a sparse, interpretable feature basis native to the model. We validate these features using visualization techniques, including activation maximization and our proposed Grad-FAM (Gradient-weighted Feature Activation Mapping), to confirm their semantic meaning. This interpretable basis is then used to guide a permanent, continuous weight-space intervention that can suppress or enhance a feature's influence. Finally, to

make this control quantifiable, we derive and validate a critical suppression threshold, $\alpha_{\text{crit}}$, providing a measure of a class's reliance on its dominant feature.

## 2 Related Work

Our work is positioned at the intersection of two primary domains: the discovery of interpretable features and the direct editing and control of model behavior. We situate our contributions at the intersection of these fields, focusing on creating a direct pathway from understanding to control.

### 2.1 Discovering Interpretable Features

A key goal of post-hoc interpretability is to make a trained model's decisions intelligible. Attribution methods, such as Grad-CAM (40), generate saliency maps that highlight influential input regions. While useful for visualization, these methods are correlational and do not expose the internal concepts the model has learned. Other approaches aim to link a model's internal components with human-understandable concepts. For example, Network Dissection (3) quantifies the semantics of individual filters by testing their alignment with a broad set of visual concepts. Similarly, TCAV (19) uses directional derivatives to measure a model's sensitivity to user-defined concepts. These methods are powerful but often rely on a pre-defined library of concepts.

Recent work in mechanistic interpretability has used SAEs to discover features in an unsupervised manner, primarily within Transformer-based models. Beyond Transformer models, SAEs have been adapted to vision and generative architectures, revealing selective remapping of visual concepts during adaptation (27). SAEs have also been applied to diffusion models for interpretable concept unlearning and sparse generative manipulation (6; 44). Recent SAE variants, including Gated SAEs (37), Top-k SAEs (10), and JumpReLU SAEs (38), have been shown to improve reconstruction fidelity and sparsity trade-offs in large models. Ongoing analyses of feature and dictionary structure provide broader insight into the properties of SAE-discovered representations (45; 46). Our work builds on this approach to feature discovery by using a SAE to automatically discover semantically meaningful features directly from the model's activations.

### 2.2 Editing and Controlling Model Behavior

Model editing techniques aim to modify a model's behavior without full retraining, typically through targeted updates. Simple interventions like ablation typically zero out neurons, or entire filters, to observe their effect on the output (24; 25; 31). However, while this provides evidence of their importance, these approaches are limited in their ability to perform fine-grained, continuous interventions, and often lack a structured representation in which such interventions can be reasoned about and controlled directly.

Most recent work using SAEs has focused on inference-time steering, where interventions involve adding a "steering vector" to a model's activations during a forward pass to influence the output (46; 51; 5; 17; 34; 43). Additional lines of work study sparse, concept-conditioned interventions at inference time, such as test-time debiasing for vision–language models (13).

Beyond steering-based approaches, other recent approaches introduce causal or counterfactual interventions (12; 11), but these remain inference-time manipulations. Complementary methods model or edit computation pathways more explicitly, including decomposing and editing predictions by modeling a network's internal computation graph (41).

Our approach is fundamentally different in its mechanism: instead of a temporary inference time adjustment to activations, we perform a permanent edit directly on the model's weights, allowing both suppression and enhancement of specific features. This is advantageous for scenarios requiring fixed, verifiable model states, eliminating the overhead of steering vectors and auxiliary modules during inference and ensuring consistent behavior across all uses of the edited model. This distinction is particularly relevant for compliance-sensitive or deployment-constrained settings.

Table 1: Comparison of related interpretability and model editing methods.

| Method Category | Examples | Primary Objective | Key Trait or Limitation |
|---|---|---|---|
| Attribution | Grad-CAM (40) | Interpret | Correlational, not causal |
| Concept-based | Dissection (3), TCAV (19) | Interpret | Requires predefined, labeled concepts |
| SAE Feature Discovery | Gated / Top-$k$ / JumpReLU SAEs (37; 10; 38) | Interpret | Basis quality varies with SAE architecture |
| Ablation | Filter Pruning (24; 25; 31) | Control | Coarse, not fine-grained |
| Causal / Counterfactual Editing | Causal Patching/Interventions (12; 11) | Control | Temporary (inference-time only) |
| Factual Editing | ROME (28), MEMIT (29) | Control | Corrective, example-dependent |
| Projection-based | INLP (39) | Control | Often requires fine-tuning |
| Architectural | CBMs (21; 50; 47) | Control | Invasive or require auxiliary concept heads |
| Computation-Path Editing | Decomposing Model Computation (41) | Interpret + Control | Requires modeling internal computation graph |
| Activation Steering | SAE Steering (46; 51; 5; 17; 34; 43), BendVLM (13) | Interpret + Control | Temporary (inference-time only) |
| **SALVE** | - | **Interpret + Control** | **Permanent edits of model-native concepts** |

Our work is related to specialized, training-free model editing techniques like ROME (28) and MEMIT (29). These methods perform corrective, example-driven edits by calculating a surgical weight update to alter a model's output for a user-provided input. In contrast, our method is feature-driven and diagnostic. Interventions are guided by general, model-native concepts discovered by the SAE. This approach enables the continuous modulation and quantitative analysis of concepts, providing a more transferable mechanism for influencing a model's overall behavior than single-instance correction.

Finally, our method differs from other concept-editing paradigms that are more invasive. For instance, Iterative Nullspace Projection (INLP) (39) removes information by projecting representations, but often requires fine-tuning to preserve model performance. Similarly, Concept Bottleneck Models (CBMs) (21) involve substantial architectural changes to incorporate a labeled concept layer. Recent work on post-hoc CBMs (50) and extensions such as stochastic CBMs (47) preserve interpretability without retraining by adding concept-heads after training. In contrast, our approach performs a localized weight edit guided by discovered features, remaining fully post-hoc and avoiding both retraining and architectural modification.

## 3 Methods

Our framework follows a three-stage "discover, validate, and control" pipeline designed to identify, understand, and intervene on a model's internal features. The stages are: (1) **Discover**, where we learn a sparse latent representation of a model's activations using an $\ell_1$-regularized autoencoder; (2) **Validate**, where we use visualization techniques to confirm the semantic meaning of the discovered features; and (3) **Control**, where we perform targeted, continuous interventions on the model's weights guided by the autoencoder's structure. We demonstrate this framework on two distinct model architectures. Our primary analysis is conducted on a ResNet-18 model fine-tuned on the Imagenette dataset (16). To demonstrate the generality and robustness across other architectures and datasets, the core stages of our methodology are successfully validated on a Vision Transformer (ViT-B/16 (8)) as well as the higher-diversity CIFAR-100 dataset (22), demonstrating that SALVE's core mechanisms transfer beyond the initial setting. Further details on both models are available in Appendix A.4. The remainder of this section details the procedures for each stage.

### 3.1 Discovering Interpretable Features

To obtain a sparse and interpretable representation of the model's internal activations, we train a linear autoencoder on the outputs of a semantically rich layer. For ResNet-18, we use the final average pooling layer, while for the Vision Transformer, we extract the representation corresponding to the `[CLS]` token after the final transformer encoder block. We use a standard reconstruction loss combined with an $\ell_1$ penalty on the latent activations to encourage sparsity. The full loss function and further architectural details are provided in Appendix A.5. To identify dominant features associated with specific output classes from this representation, we compute the class-conditional mean of the latent activations, $\mu_k = \frac{1}{|C_k|} \sum_{n \in C_k} Z_n$, where $C_k$ is the set of samples for class $k$ and $Z_n$ is the latent activation vector for a single sample $n$. Analyzing $\mu_k$ reveals which features are strongly associated with a particular class, providing a basis for targeted interventions.

## 3.2 Validating and Controlling Features

To validate the semantic content of the discovered features, we use two complementary visualization techniques. We employ activation maximization (33), extending it from traditional applications on individual neurons or filters to our discovered latent features to synthesize the abstract concept a feature represents. To ground feature activations in specific input images, we introduce Grad-FAM (Gradient-weighted *Feature* Activation Mapping), an adaptation of Grad-CAM (Gradient-weighted *Class* Activation Mapping). While standard Grad-CAM generates saliency maps for a class logit, our method repurposes this logic for a specific latent feature, providing a direct visual link between the feature and the input regions that activate it. See Appendix A.6 for the full derivation and implementation details.

Having validated that the features represent meaningful semantic concepts, we investigate their causal role by using the autoencoder's decoder matrix, $D \in \mathbb{R}^{M \times d}$, to guide a permanent, continuous edit to the weights of the model. Each column of this matrix, $D[:, l]$, represents the "direction" of a latent feature $l$ in the original activation space. We edit the final-layer weights $w_{ij}$ as follows:

$$w'_{ij} = w_{ij} \cdot \max(0, \, 1 \pm \alpha \cdot |c_j|), \tag{1}$$

where $c_j = D[j, l]$ is the contribution of the latent feature to the original feature $j$, and $\alpha$ controls the intervention strength. The $\pm$ symbol indicates that we can either enhance (+) or suppress (-) the feature's influence, depending on the desired intervention. Edits are performed along a feature direction, and we obtain class-targeted effects by selecting a feature dominant for that class. This feature-guided modulation differs from traditional ablation, as it allows for fine-grained control over the model's learned concepts while preserving the network's structure. Because the edit acts multiplicatively on the learned weights rather than additively on activations, its effect remains input-dependent and interacts with the sample's actual feature composition $x$ (i.e., the modulation combines with the activation vector rather than overriding it). This distinction from additive steering is what enables per-sample diagnostics, including the critical suppression threshold $\alpha_{\text{crit}}$ introduced in the next paragraph.

To quantify a class's reliance on a feature, we first define the feature-perturbed logit, $z'_i(\alpha)$, as a function of the intervention strength $\alpha$:

$$z'_i(\alpha) = \sum_j w_{ij} \cdot \max(0, \, 1 - \alpha |c_j|) \cdot x_j, \tag{2}$$

where $x_j$ is the $j$-th component of the activation vector from the penultimate layer. This gives the class logit under feature perturbation, excluding the global class bias term $b_i$ so that we measure the feature's influence independently of baseline class predisposition. We define the critical suppression threshold, $\alpha_{\text{crit}}$, as the value of $\alpha$ for which $z'_i(\alpha) = 0$. Intuitively, $\alpha_{\text{crit}}$ is the intervention strength required to completely suppress the logit contribution attributable to this feature direction. In the regime of weak perturbations ($\alpha < 1/|c_j|$), a linear approximation yields the analytical estimate:

$$\alpha_{\text{crit}}^{(n)} \approx \frac{z_i^{(n)}}{R_i(\mathbf{x}^{(n)})}, \tag{3}$$

where $z_i^{(n)} = \sum_j w_{ij} x_j^{(n)}$ is the original (bias-free) class logit for sample $n$, and $R_i(\mathbf{x}^{(n)}) = \sum_j |c_j| |w_{ij} x_j^{(n)}|$ quantifies the logit's sensitivity to suppression along the latent feature direction. While the analytical estimate provides a lower bound, we also compute $\alpha_{\text{crit}}^{(n)}$ numerically. The full derivation and further details are provided in Appendix A.7.

## 4 Results

We validate our framework through two main analyses. First, we identify and visualize the discovered latent features to confirm they represent meaningful semantic concepts. Second, we perform targeted weight-space interventions to demonstrate precise control over the model's output. Together, these analyses demonstrate our framework's ability to link interpretation directly to intervention.

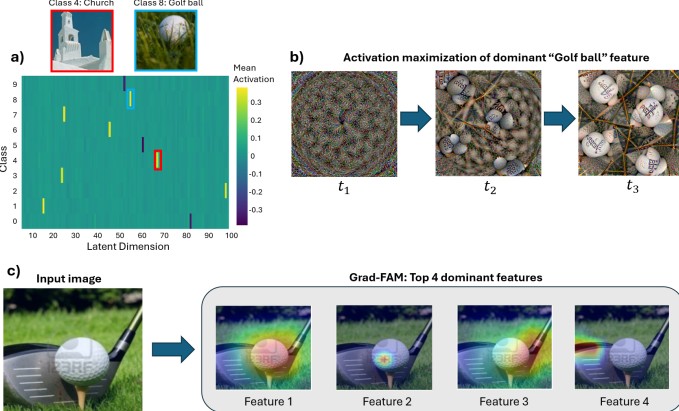

Figure 1: Validation of discovered features for the ResNet-18 model. **(a)** Average latent feature activations across classes, showing a sparse, class-specific basis. **(b)** Activation maximization of the "golf ball" feature. The image evolves from random noise ($t_1$), through emergence of circular shapes ($t_2$), to the final image clearly exhibiting golf ball characteristics ($t_3$). **(c)** Grad-FAM visualizations grounding the top-4 dominant features for a sample "golf ball" image.

### 4.1 LATENT FEATURES ENCODE SEMANTIC CONCEPTS

Our first objective is to validate that the features discovered by the SAE correspond to meaningful visual concepts. We find that the SAE successfully learns a class-specific feature basis. As shown in the average feature activations in Figure 1a, the ResNet-18 representations have a sparse, dominant structure where a single feature is strongly associated with a single class. While these dominant features define a class, the less active features often represent finer-grained concepts shared across classes, which we explore further in Section 4.3.

To understand what these features represent abstractly, we first use activation maximization (see Appendix A.6.1 for further details). Figure 1b shows this for the "golf ball" feature, where the optimization reveals objects with the characteristic texture of a golf ball. To further visualize how these features are grounded in specific inputs, we use our proposed Grad-FAM method (see Appendix A.6 for the full derivation and implementation details). Figure 1c shows an example for a golf ball image. The dominant "Feature 1" provides a high-level concept map for the "golf ball" class, whereas less dominant features correspond to granular sub-concepts. For example, "Feature 2" activates on the ball's surface texture, while "Features 3 and 4" highlight different parts of the golf club.

To confirm these findings are not an artifact of the convolutional architecture, we replicate our core validation analyses on the Vision Transformer (ViT). We find that the ViT also learns a sparse, class-specific feature basis and that Grad-FAM successfully localizes these features to semantically relevant image regions (see Appendix A.11.1). We additionally evaluate feature extraction on CIFAR-100 to test dataset-level robustness. Despite the higher class granularity, we observe a structured and relatively sparse latent representation (see Appendix A.10). Dominant features still emerge but exhibit more cross-class sharing than in Imagenette, an expected consequence of the dataset's breadth and the use of a simple linear $\ell_1$-regularized SAE. Nevertheless, the discovered features support the same diagnostic and intervention analyses presented in the main experiments. Together, these results confirm that the discovered features provides a valid basis for targeted interventions.

### 4.2 CONTROLLING CLASS PREDICTIONS VIA FEATURE EDITING

Having validated the semantic meaning of the features, we now evaluate their causal role by performing permanent model edits. We first focus on the dominant, class-specific features. A qualitative case study on an ambiguous, out-of-distribution image (not part of Imagenette) containing both a "golf ball" and a "church" demonstrates the precision of our method. This

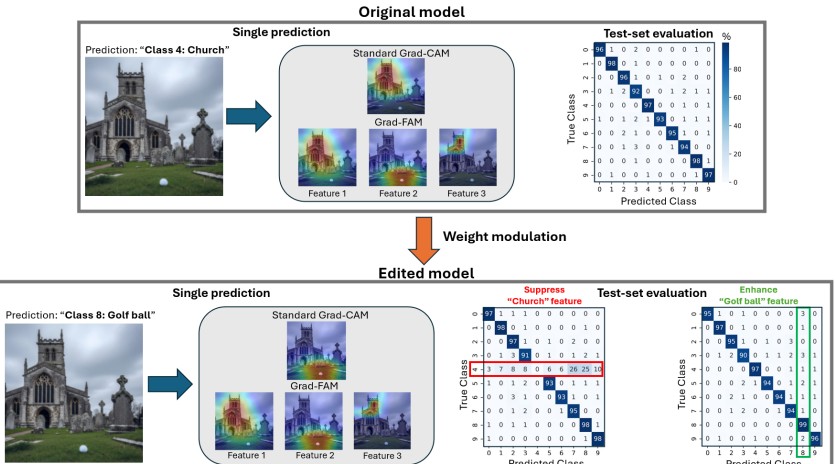

Figure 2: (Left) A qualitative case study where suppressing the "Church" feature or enhancing the "Golf ball" feature successfully flips the model's prediction for an ambiguous image. (Right) Quantitative validation on the test set, showing minimal off-target effects

experiment is illustrated under "Single prediction" in Figure 2. The original model predicts "Church", and as a comparative illustration we calculate both the standard Grad-CAM as well as Grad-FAM saliency maps. The Grad-CAM indicates that the model primarily focuses on the tower of the church to make its prediction. Using Grad-FAM, we confirm that the dominantly activated feature corresponds to the church structure (similar to the Grad-CAM), but it also reveals other features activated by the golf ball (Feature 2) and by the church tower (Feature 3).

We then perform two interventions. First, suppressing the dominant "Church" feature predictably flips the classification to "Golf ball". Conversely, enhancing the "Golf ball" feature achieves the same outcome. Post-edit Grad-CAMs confirm the model's attention shifts accordingly in both cases, from focusing primarily on the church tower to the golf ball instead. To validate this effect quantitatively, we evaluate the model on the entire Imagenette test set. The confusion matrices in Figure. 2 for the edited model show two key results. First, suppressing the "Church" feature disables the model's ability to recognize that class, reducing its accuracy to near zero. Second, enhancing the "Golf ball" feature maintains its near-perfect accuracy, indicating the edit is stable and does not degrade performance on well-learned and robust class representations. Critically, for both interventions, performance on the other classes remains relatively unaffected. This confirms that the learned class-dominant features enable precise, modular control, allowing for targeted interventions with minimal off-target effects. While this example targeted a specific class, the same methodology was successfully applied to the other classes in the dataset.

To validate the architectural robustness of this control method, we replicate these class-suppression experiments on the Vision Transformer and achieve a similar degree of precise, modular control over its predictions (see Appendix A.11.2). To assess robustness across datasets, we perform additional experiments on CIFAR-100 (see Appendix A.10). Suppressing a class's dominant feature reliably reduces its recognition accuracy, mirroring the behavior observed on Imagenette. Despite the higher class granularity, off-target effects remain limited, though higher than in Imagenette due to the increased visual diversity, and the edits exhibit the same qualitative pattern of targeted suppression. These results indicate that feature-guided weight edits generalize even when the underlying feature basis exhibits greater cross-class sharing.

To contextualize our results, we include two complementary baselines: (i) ROME (28), a well-known method for example-driven factual editing in large language models, adapted here for vision models as a permanent weight-editing approach; and (ii) SAE-based Activation Steering, an inference-time intervention leveraging the same latent feature space as our

method. Both baselines achieve similar outcomes on the class suppression task, reducing the target class accuracy to near zero with minimal off-target effects (see Appendix A.8 and Appendix A.9 for details). Despite this similarity in outcome, the underlying mechanisms differ substantially. ROME performs a rank-one weight update based on a single-sample key, while Activation Steering applies a uniform additive offset along a concept direction during inference. In contrast, SALVE offers unique benefits: permanent edits without inference overhead, systematic control over multiple latent concepts, and quantitative diagnostics such as $\alpha_{\text{crit}}$. These distinctions make SALVE competitive for global interventions while offering practical advantages for fine-grained interpretability and reliability assessments. This feature-driven approach underpins the capabilities explored in the following sections: continuous modulation of concept influence, quantitative diagnostics such as $\alpha_{\text{crit}}$, and nuanced cross-class edits that reveal the structure and brittleness of learned representations.

### 4.3 Intervening on Cross-Class Features

To demonstrate our framework's ability to edit fine grained concepts shared across classes, we identify a "Tower Feature", which is frequently activated by images of both churches (Class 4) as well as petrol pumps (Class 7) containing tower-like structures. A selection of the top-activating images from the test set, shown in Figure 3a, confirm this cross-class association. We then perform symmetrical interventions on this feature (Figure 3b). Suppressing the feature reduces accuracy for "Petrol Pump" while leaving the "Church" class largely unaffected. Conversely, enhancing the feature increases the accuracy for "Petrol Pump". This differential impact suggests that the model's classification of certain older, column-shaped petrol pumps is highly reliant on the "Tower Feature". In contrast, the "Church" class appears more robust due to a richer set of redundant features (e.g., stained glass, steeples, etc.). Notably, these interventions also revealed a subtle feature entanglement.

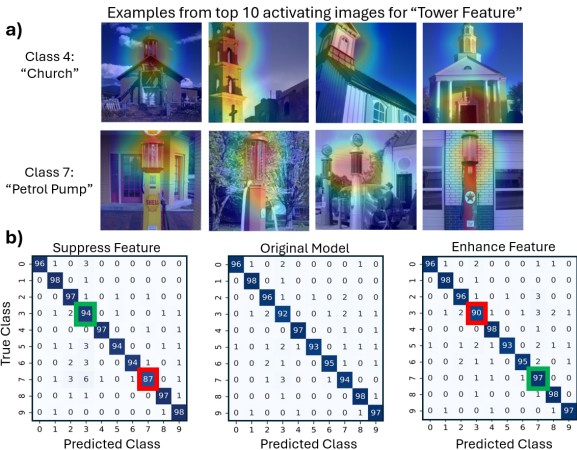

Figure 3: a) Validation of the "Tower Feature", showing example top-activating images from the test set. b) Effect of suppressing and enhancing this feature on model predictions, where "red" and "green" corresponds to decreasing or increasing class accuracy, respectively

Suppressing the "Tower Feature" slightly increased accuracy for "Chain Saw" (Class 3), while enhancing it caused a corresponding decrease. This consistent, inverse effect, suggests that the model has learned a spurious negative correlation, causing the "Tower Feature" to act as an inhibitor for the "Chain Saw" classification. This illustrates a nuanced relationship that would be difficult to uncover without such targeted interventions.

### 4.4 Quantifying Intervention Sensitivity

Having established that interventions are effective, we now quantify their sensitivity to the scaling factor $\alpha$. Systematically varying $\alpha$ allows us to probe the causal importance of each feature and measure a class's reliance on it. Figure 4a shows a characteristic suppression

curve for the "Church" class. As $\alpha$ increases, its accuracy remains stable before dropping sharply past a critical threshold, while performance on other classes remains high, confirming the targeted nature of the edit. This drop in accuracy corresponds to a reallocation of the model's predictions. As the dominant feature is suppressed, the model's confidence is redistributed among the other classes (Figure 4b)

To evaluate the sensitivity of the learned latent basis on the class intervention, we performed experiments across multiple initializations and training runs (see Appendix A.12.1). Crucially, the results are consistent across multiple SAE training runs, as indicated by the narrow shaded regions in the curves of Figure 4, representing the standard deviation across $n = 10$ different realizations. This demonstrates that while the specific basis vectors learned by the SAE may vary, the effect of our intervention on the model's underlying concepts is stable.

We then extend this analysis by computing suppression curves for the dominant feature of all classes. As these results proved robust to different autoencoder initializations, we simplify the subsequent analysis by presenting results from a single realization. We observe a similar suppression behavior across all classes for the ResNet-18 model (Fig. 5a), with variations in the threshold suggesting that each class relies on its dominant feature to a different degree. To confirm that this behavior was not an artifact of the convolutional architecture, we replicate the same analysis on the Vision Transformer. The ViT exhibited the same characteristic suppression curves, indicating that this is a general property of the intervention method (see Appendix A.11.3). As the results are robust to the stochasticity of the SAE training process, this indicates that the observed differences in class sensitivity stem mainly from the base model's intrinsic feature representations. In addition, the properties of the discovered feature basis are also dependent on the SAE's training objective (e.g., the sparsity coefficient $\lambda_1$). Therefore, the observed heterogeneity between classes is a product of both the backbone's intrinsic structure and the specific feature decomposition learned by the SAE. Further details assessing the robustness of our experiments are provided in Appendix A.12.

### 4.4.1 CRITICAL SUPPRESSION STRENGTH: $\alpha_{\mathrm{crit}}$

To quantify this class-level feature dependency, we calculate the critical suppression threshold, $\alpha_{\mathrm{crit}}$ (derivation and implementation details are found in Appendix A.7). This metric estimates, on a per-sample basis, the precise intervention strength required to reduce the feature contribution to the logit for the correct class to zero, providing a granular measure of how strongly the model relies on a specific feature for a given input. We compute $\alpha_{\mathrm{crit}}$ in two ways: an analytical estimate based on the linear approximation (given by Eq.3), and a numerical calculation (given by Eq.2). To validate both, we compare their distributions to an empirical threshold, $\alpha_{50\%}$. It is important to note the conceptual distinction between

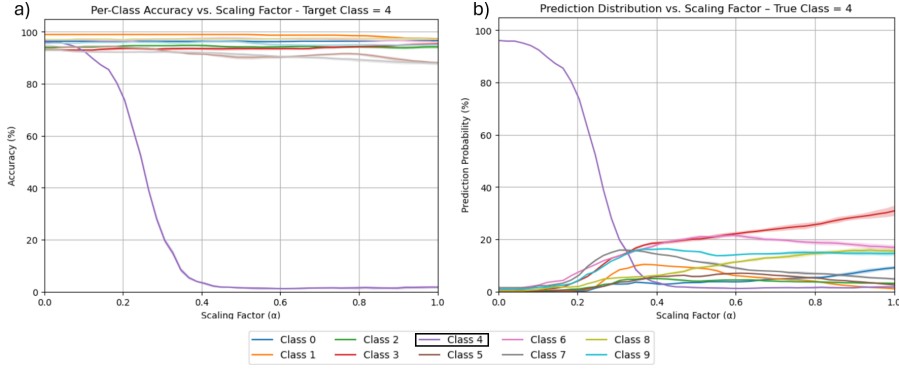

Figure 4: Suppression sensitivity for "Church" (Class 4). (a) Per-class accuracy vs. intervention strength $\alpha$. (b) Distribution of predictions for images of the target class, showing how confidence is reallocated as the feature is suppressed. Shaded regions indicate the standard deviation across 10 SAE initializations.

these metrics. The analytical and numerical $\alpha_{\mathrm{crit}}$ are per-sample measures that identify the intervention strength required to drive the correct class's feature contribution to the logit to zero, representing a point where the feature's evidence for that class is entirely suppressed. In contrast, the empirical $\alpha_{50\%}$ is a population-level metric that marks the point where accuracy on the class falls to 50%. This happens when, for a typical sample, the logit for the correct class has dropped just enough to be surpassed by a competing logit. Figure 5b presents this comparison for the ResNet-18 model. As expected, the analytical estimate provides a lower bound on the critical intervention strength. The numerically calculated $\alpha_{\mathrm{crit}}$, in turn, aligns well with both this lower bound and the empirical estimate, and its distribution also captures a wider range of values for samples where the linear approximation is less valid.

To confirm the architectural generality of the $\alpha_{\mathrm{crit}}$ metric, we replicate this analysis also on the Vision Transformer. Here, we find two key differences compared to the ResNet-18 model. First, the analytical estimate of $\alpha_{\mathrm{crit}}$ represents a much more conservative lower bound. Second, we observe a larger discrepancy between the numerically calculated $\alpha_{\mathrm{crit}}$ (representing total evidence loss) and the empirical $\alpha_{50\%}$ (representing a typical flip of predictions). We hypothesize that these observations can be attributed to the ViT's architectural properties. Recent research has shown that ViTs learn a "curved" and non-linear representation space (20). This effect, combined with the ViT's tendency to produce less confident, more distributed logits (30), means a smaller intervention is required to be surpassed by a competitor. This widens the gap between the point of prediction failure ($\alpha_{50\%}$) and total evidence loss ($\alpha_{\mathrm{crit}}$), an effect less pronounced in the more linear representations of the ResNet model. (See Appendix A.11.3 for results and further discussions).

Analyzing the distribution of $\alpha_{\mathrm{crit}}$ enables both class-level and sample-level diagnostics, allowing us to pinpoint fragile representations that may be susceptible to adversarial perturbations and to guide targeted strategies for improving robustness.

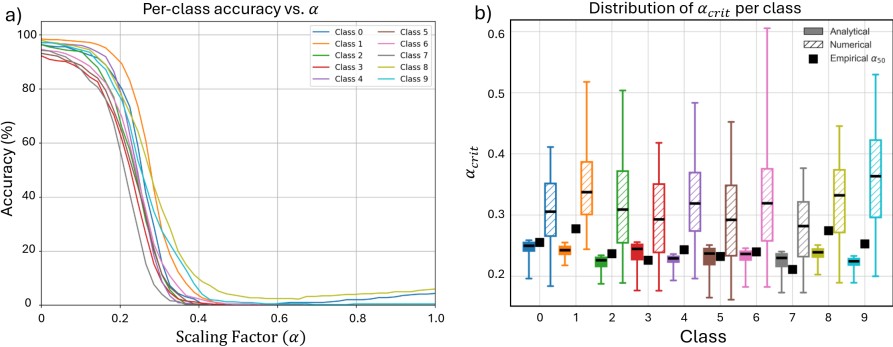

Figure 5: **(a)** Per-class accuracy vs. the intervention strength $\alpha$. **(b)** Comparison of the critical suppression threshold estimates. The distributions of the per-sample analytical (filled box) and numerical (hatched box) $\alpha_{\mathrm{crit}}$ are shown as boxplots. The central line indicates the median, the box spans the interquartile range (25th to 75th percentile), and the whiskers extend to the 5th and 95th percentiles. The empirical threshold ($\alpha_{50\%}$) is overlaid as a square marker.

## 5 Limitations and Future Work

While our framework provides a complete pipeline for discovering, validating, and controlling a model's internal features, this study was intentionally scoped as a mechanistic case analysis on basic datasets and well-understood architectures. This controlled setting allowed us to isolate and evaluate the core principles of SALVE, but broader validation remains an important next step and motivates the future directions discussed below.

A primary direction for future work is to scale our validation across a broader range of models and datasets. The results of our study, supported by recent findings that sparse autoencoders

effectively uncover semantic concepts in Vision Transformers (27; 34; 43), provide a strong foundation for this expansion. In our additional CIFAR-100 experiments, we observe that the overall methodology still generalizes, but the discovered feature basis exhibits greater cross-class sharing, reflecting the limits of a simple linear $\ell_1$ SAE in high-diversity settings. This highlights the need for evaluating more recent SAE variants, such as Gated, Top-$k$, or JumpReLU SAEs, to improve reconstruction fidelity and feature separation when scaling to larger models and more complex datasets. Investigating how properties like feature modularity and intervention efficacy vary between different architectures, and extending this comparative analysis to larger models, more complex datasets, and other data modalities like natural language processing, represents crucial next steps.

Furthermore, we found that intervention effectiveness depends on both the backbone model's training dynamics (e.g., batch size) and the parameters used to train the SAE (e.g., $\ell_1$ regularization strength) (see Appendix A.12.2 for a more detailed discussion). This indicates a crucial link between training procedures and post-hoc controllability. Future work should investigate how to co-design training and intervention methods to produce models that are not only performant but also inherently more editable. Our work also opens several avenues for a deeper feature-level analysis. While we identified concrete instances of feature entanglement (e.g., the "Tower Feature", as illustrated in Figure 3a), future work could incorporate formal disentanglement metrics to provide a more rigorous evaluation. Additionally, a powerful extension would be to adapt our weight modulation technique to deeper layers of the model. Such a method could enable more fundamental edits to a model's core feature representations, moving from modifying how concepts are combined to changing how they are formed.

Finally, our $\alpha_{\text{crit}}$ metric suggests a link between feature reliance and model robustness. Systematically investigating if low-$\alpha_{\text{crit}}$ features and samples correlate with known adversarial vulnerabilities represents another promising direction for future work.

## 6 Conclusions

This work introduced SALVE, a framework for interpreting and controlling neural networks through a unified *discover–validate–control* pipeline. By training a sparse autoencoder (SAE) on a model's internal activations, we extracted a basis of model-native features that capture both dominant, class-defining concepts and fine-grained representations shared across classes. We validated these concepts using activation maximization to visualize their abstract form and our proposed Grad-FAM method to ground their presence in specific input regions.

Our primary contribution is demonstrating that this interpretable feature basis can guide precise, permanent, and post-hoc edits to model weights. SALVE enables continuous modulation of concept influence and supports nuanced, cross-class interventions. To make this control quantifiable, we introduced the critical suppression threshold, $\alpha_{\text{crit}}$, a per-sample metric that measures reliance on specific latent features. This provides a foundation for future work on diagnosing brittle representations and adversarial vulnerabilities.

To assess the effectiveness of SALVE, we benchmarked it against two complementary baselines: ROME, a rank-one weight-editing method adapted from language models, and SAE-based Activation Steering, an inference-time intervention leveraging the same latent feature representation. While all three methods achieve similar outcomes, SALVE offers key advantages: permanent edits without inference overhead, systematic control over multiple latent concepts, and quantitative diagnostics such as $\alpha_{\text{crit}}$, enabling fine-grained interpretability and robustness auditing.

Our methodology was validated on representative architectures (ResNet-18 and ViT-B/16), demonstrating the feasibility of interpretable control in both convolutional and transformer-based models. By providing a principled approach to uncover, validate, and manipulate a model's internal concepts, SALVE offers a foundation for future work toward more transparent, robust, and reliable AI systems, and opens several promising directions for continued research.

## 7 REPRODUCIBILITY STATEMENT

The methodology, derivations, parameters, and implementation details are described in the Appendix A). A comprehensive and fully documented GitHub repository will be made publicly available upon publication.

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

# A Appendix

## A.1 LLM Usage

LLM-based tools were utilized for enhancing the writing quality and clarity of the paper, and for providing support in writing parts of the analysis code. Specifically, Gemini 2.5 Pro was used for grammar and style text polishing and coding support.

The authors maintain full responsibility for the content, accuracy, and conclusions presented in this paper.

## A.2 Compute Resources

All experiments in this paper were performed on a single local workstation, and can be reproduced on a standard modern laptop with GPU acceleration.

## A.3 Fine-tuning of Pre-trained Backbone Models

We initialized our feature extraction backbone using standard, pre-trained architectures. We used a ResNet-18 model (14) and a Vision Transformer (ViT-B/16 (8)) with a patch size of 16x16. Both models were pre-trained on the ImageNet dataset (7) and loaded via established PyTorch libraries (35).

Because the Imagenette dataset (16) contains 10 classes and CIFAR-100 (22) contains 100 classes, the final classification layer of each model was replaced to match this number of outputs. For ResNet-18, this involved replacing the `fc` layer, and for the ViT, the `head` layer.

Both models were fine-tuned using the Adam optimizer with categorical cross-entropy loss and a batch size of 16. For the Imagenette analysis, the ViT was fine-tuned for 5 epochs with a learning rate of $10^{-5}$, and the ResNet-18 for 10 epochs with a learning rate of $10^{-4}$. We additionally performed sensitivity experiments on the effect of varying the backbone's batch size (see Section A.12.2). For our CIFAR-100 validation experiments, we restricted our analysis to the ResNet-18 backbone, using the same optimizer, batch size, learning rate, and number of epochs as for Imagenette.

## A.4 Fine-tuning of Pre-trained Backbone Models

## A.5 Autoencoder Definition and Training

To learn a sparse, interpretable representation of the backbone models' features, we trained an autoencoder to reconstruct high-dimensional activation vectors extracted from a specific layer in the backbone. The activations were extracted using forward hooks in PyTorch. The target layer was chosen to be a late-stage, semantically rich layer just before the final classification head.

- For ResNet-18, we extracted the 512-dimensional feature vector from the output of the final average pooling layer (`avgpool`).
- For the ViT, we extracted the 768-dimensional representation corresponding to the `[CLS]` token after the final transformer encoder block.

This process yielded a matrix of activations $A \in \mathbb{R}^{N \times M}$, where $N$ is the number of samples and $M$ is the feature dimensionality of the chosen layer. The autoencoder's encoder maps these activations to a latent space $Z \in \mathbb{R}^{N \times d}$, and the decoder reconstructs them as $\hat{A} \in \mathbb{R}^{N \times M}$.

The architecture of the autoencoder is defined as follows:

```
class SparseAutoencoder(nn.Module):
    def __init__(self, input_dim, hidden_dim):
        super(SparseAutoencoder, self).__init__()
        self.encoder = nn.Linear(input_dim, hidden_dim)
        self.decoder = nn.Linear(hidden_dim, input_dim)
```

```
        nn.init.xavier_uniform_(self.encoder.weight)
        nn.init.xavier_uniform_(self.decoder.weight)

    def forward(self, x):
        encoded = self.encoder(x)
        decoded = self.decoder(encoded)
        return encoded, decoded
```

The model was trained to minimize the reconstruction loss combined with an $\ell_1$-regularization term to promote sparsity:

$$\mathcal{L} = \|A - \hat{A}\|_F^2 + \lambda_1 \|Z\|_1. \tag{4}$$

This sparsity encourages modularity, such that any given input activates only a few latent dimensions. The models used in this study have low latent dimensionality, making sparse feature extraction with good reconstruction fidelity tractable with a basic SAE with $\ell_1$-regularization. Scaling to larger datasets or more complex architectures will likely benefit from SAE variants more widely used in large-scale LLM interpretability (e.g., JumpReLU, gated, top-k).

The autoencoders for both models were trained for 1000 epochs using an Adam optimizer and a step decay learning rate scheduler (reducing LR by a factor of 0.8 every 200 epochs). The ResNet-18 SAE used a learning rate of $10^{-3}$, batch size of 32, and $\lambda_1 = 10^{-3}$, while the ViT-B/16 SAE used a learning rate of $10^{-4}$, batch size of 16, and $\lambda_1 = 10^{-2}$.

The training loss curves and examples of the reconstructed activations are illustrated in Figure 6.

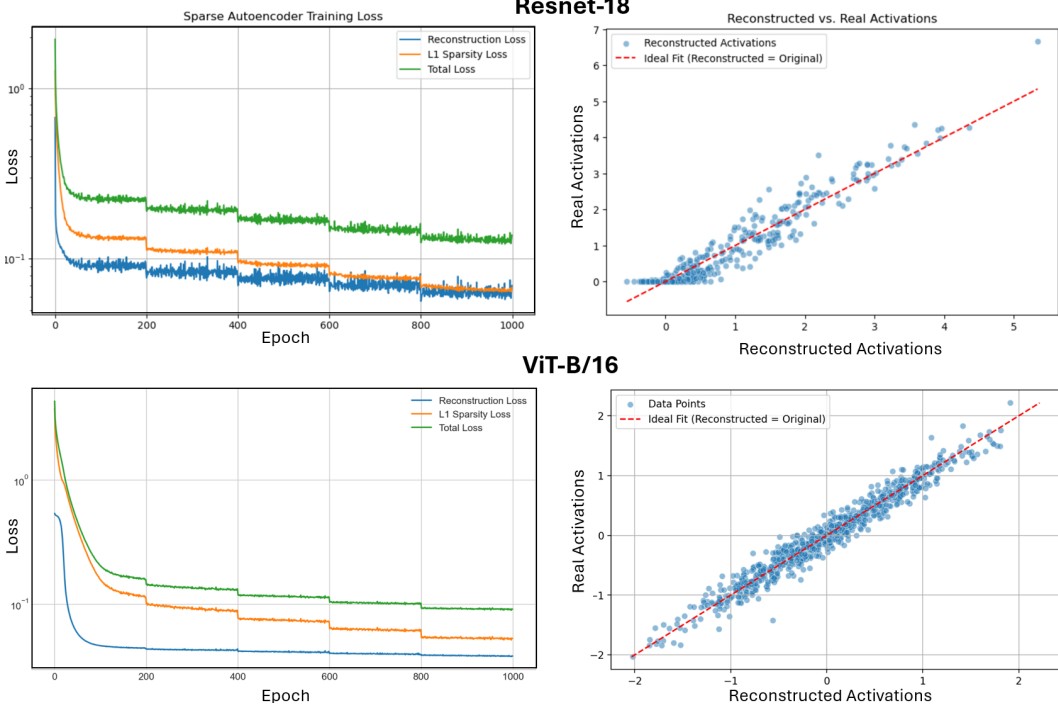

Figure 6: Autoencoder training and reconstruction performance for the ResNet-18 and ViT-B/16 backbones. **Left**: Training curves for the Sparse Autoencoders, showing total loss, reconstruction loss, and L1 sparsity loss (log-scaled) per epoch. **Right**: A comparison of the autoencoder's reconstructed feature activations with the original ones, where the red dashed line indicates perfect reconstruction ($\hat{A} = A$).

## A.6 VISUALIZATION OF LATENT FEATURES

### A.6.1 ACTIVATION MAXIMIZATION

To visualize the abstract concept encoded by a feature, we employ activation maximization (33), extending it from traditional applications on individual neurons or filters to the discovered latent features. The method synthesizes an image from random noise by optimizing its pixels to maximally activate a chosen feature, revealing the visual pattern it has learned to detect.

In this procedure, we optimize an input image $x \in \mathbb{R}^{C \times H \times W}$, where $C$ denotes the number of channels and $H$, $W$ represent the image dimensions. The objective is to maximize the activation $\phi_l(x)$ of a target latent feature $l$, previously identified as dominant for a particular class. The optimization objective is defined as:

$$\underset{x}{\text{maximize}} \quad \phi_l(x) - \lambda_{\text{L2}} \cdot \|x\|_2^2 - \lambda_{\text{TV}} \cdot \text{TV}(x), \tag{5}$$

where $\phi_l(x)$ is the activation of the $l$-th latent feature given input $x$, $\|x\|_2^2$ penalizes large pixel values, and $\text{TV}(x)$ is the total variation loss. This encourages smoother and more natural visual structures by discouraging high-frequency noise and sharp changes between neighboring pixels, and is formally defined as:

$$\text{TV}(x) = \sum_{c=1}^{C} \sum_{i=1}^{H-1} \sum_{j=1}^{W-1} \sqrt{(x_{c,i+1,j} - x_{c,i,j})^2 + (x_{c,i,j+1} - x_{c,i,j})^2}, \tag{6}$$

The regularization parameters $\lambda_{\text{L2}}$ and $\lambda_{\text{TV}}$ control the strength of the respective penalties, helping ensure that the optimized image is visually coherent, rather than adversarial or noisy (33).

To further increase the robustness of the visualization, we apply a sequence of stochastic image transformations at each optimization step. Specifically, we apply constant padding (16 pixels, value 0.5), random jittering (8 pixels), random scaling $[-\delta\%, +\delta\%]$, and random rotation $[-\theta, +\theta]$. After augmentation, a center crop restores the input to a fixed resolution of $224 \times 224$ pixels. These transformations, inspired by prior work on robust feature visualization (9; 49), act as implicit regularizers that prevent overfitting to high-frequency artifacts and encourage the emergence of semantically meaningful patterns.

The complete procedure is given by the following steps:

1. Initialize the input image $x$ with random noise.

2. For a fixed number of iterations:

    (a) Apply the sequence of image transformations (padding, jittering, scaling, rotation, center cropping).

    (b) Forward the transformed image through the network and encoder to obtain the latent activations.

    (c) Evaluate the objective function combining target activation, total variation loss, and $L_2$ regularization.

    (d) Update $x$ using gradient ascent to maximize the objective function.

### A.6.2 GRAD-FAM IMPLEMENTATION

While standard Grad-CAM is designed to produce saliency maps that highlight image regions influencing the final prediction for a specific *class*, Grad-FAM (Gradient-weighted Feature Activation Map), repurposes this logic to visualize which parts of an image are responsible for activating a specific *latent feature* learned by the SAE.

The core modification lies in the target of the gradient calculation. In standard Grad-CAM, the heatmap is generated by weighting the feature maps $F^k$ from a target layer by importance weights $\beta_k^c$. These weights are calculated by global average pooling the gradients of the score

for a class $c$, $y^c$, with respect to the feature maps. Denoting the number of pixels in the feature map as $P$, the importance weight for a feature map $k$ is given by:

$$\beta_k^c = \frac{1}{P} \sum_i \sum_j \frac{\partial y^c}{\partial F_{ij}^k} \tag{7}$$

In Grad-FAM, we replace the class score $y^c$ with the activation value of a latent feature, $\phi_l$, from our learned sparse representation. The importance weight $\beta_k^l$ is therefore calculated based on the gradient of this specific latent feature's activation:

$$\beta_k^l = \frac{1}{P} \sum_i \sum_j \frac{\partial \phi_l}{\partial F_{ij}^k} \tag{8}$$

Here, $F^k$ represents the $k$-th channel of the activation tensor from the target layer. While the formulation is model-agnostic, its application requires handling architectural differences. For a CNN like ResNet, $F^k$ is an inherently spatial 2D feature map. A Vision Transformer, however, outputs a 1D sequence of tokens. To apply the same spatially-based pooling, these tokens must be reshaped back into a 2D grid. This transformation is achieved by first discarding the non-spatial [CLS] token from the sequence, then reshaping the remaining $N$ patch tokens into their original $\sqrt{N} \times \sqrt{N}$ grid. This process correctly restores the visual layout because the token sequence preserves the original raster-scan order of the image patches.

The final heatmap, $L^l$, is produced by taking the absolute value of the weighted combination of feature maps. This differs from the conventional Grad-CAM method, which applies a ReLU. This choice is motivated by the design of our latent feature representation ($\phi_l$), which can take on both positive and negative values to represent concepts. Since a strong influence in either direction is a meaningful signal of importance, we take the absolute value to ensure our heatmap highlights all regions that significantly contribute to the feature's final activation.

$$L^l = \left| \sum_k \beta_k^l F^k \right| \tag{9}$$

To implement this, the forward pass must be explicitly chained from the base model through the autoencoder's encoder to establish a continuous computational graph for backpropagation. We make use of PyTorch hooks to capture the necessary intermediate data during this single forward and backward pass. The full procedure is summarized in the following pseudocode.

```
function Generate-Grad-FAM(model, autoencoder, image, target_layer, feature_index):
    # 1. Register hooks on target_layer to capture feature maps and gradients
    register_hooks(target_layer, forward_hook, backward_hook)

    # 2. Forward pass to get the latent score
    internal_activations = model.get_internal_activations(image)
    latent_vector = autoencoder.encoder(internal_activations)
    target_score = latent_vector[feature_index]

    # 3. Backward pass from the latent score
    target_score.backward()

    # 4. Generate the heatmap
    feature_maps = captured_feature_maps
    gradients = captured_gradients

    # For ViT, reshape token-based maps/gradients to a spatial grid
    if is_transformer(model):
        feature_maps = reshape_transform(feature_maps)
```

```
        gradients = reshape_transform(gradients)

    pooled_gradients = global_average_pool(gradients)
    weighted_maps = weight_feature_maps(feature_maps, pooled_gradients)
    # Take absolute value to get magnitude of influence
    heatmap = abs(sum(weighted_maps))
    normalized_heatmap = normalize(heatmap)

    return normalized_heatmap
```

By backpropagating from an intermediate latent feature instead of a final class logit, Grad-FAM shifts the focus from standard input attribution to the spatial localization of internal concepts. The resulting saliency maps thus provide a direct, visual link between the abstract features learned by the autoencoder and their manifestation in the input data.

## A.7   CRITICAL SUPPRESSION THRESHOLD

### A.7.1   DERIVATION AND PROCEDURE: ANALYTICAL METHOD

Here, we provide the full derivation for the critical suppression threshold, $\alpha_{\text{crit}}$, introduced in Section 3.2. The goal is to analyze how our weight modulation affects the feature contribution to the logit for a class $i$.

Let $\mathbf{x}$ be the activation vector from the penultimate layer. The feature-perturbed logit, $z_i'(\alpha)$, after applying weight suppression with scaling factor $\alpha$, is then given by:

$$
\begin{aligned}
z_i'(\alpha) &= \sum_j w_{ij}' x_j \\
&= \sum_j w_{ij} \cdot \max(0,\ 1 - \alpha\,|c_j|) \cdot x_j,
\end{aligned}
\tag{10}
$$

where $w_{ij}$ is the original weight and $c_j$ is the contribution of the target latent feature to the original feature $j$. This definition isolates the direct contribution to the logit from the model features, excluding the global class bias term.

For small values of $\alpha$, we can use the linear approximation: $\max(0, 1 - y) \approx 1 - y$. This approximation is valid under the condition that $\alpha < 1/|c_j|$ for all relevant components $j$ (see A.7.3 for validation of this approximation).

Applying this, we get:

$$
\begin{aligned}
z_i'(\alpha) &\approx \sum_j w_{ij}(1 - \alpha|c_j|)x_j \\
&= \sum_j w_{ij}x_j - \alpha \sum_j |c_j|w_{ij}x_j \\
&= z_i - \alpha\, R_i(\mathbf{x}),
\end{aligned}
\tag{11}
$$

where $z_i = \sum_j w_{ij}x_j$ is the original feature contribution to the logit, and we define the sensitivity term as:

$$
R_i(\mathbf{x}) = \sum_j |c_j|w_{ij}x_j
\tag{12}
$$

This term, $R_i(\mathbf{x})$, quantifies how sensitive the logit $z_i$ is to suppression along the direction of the chosen latent feature. We define the critical threshold as the value of $\alpha$ where the feature contribution of the logit is driven to zero, which gives the per-sample threshold for sample $n$:

$$
z_i^{(n)} - \alpha_{\text{crit}}^{(n)} R_i(\mathbf{x}^{(n)}) = 0 \quad \Longrightarrow \quad \alpha_{\text{crit}}^{(n)} = \frac{z_i^{(n)}}{R_i(\mathbf{x}^{(n)})}
\tag{13}
$$

Our model is trained for multi-class classification using a standard cross-entropy loss function. Because this loss function in PyTorch internally applies a log-softmax operation, the model's final layer is correctly designed to output raw, unbounded logits. For the purpose of our

$\alpha_{\text{crit}}$ derivation, we thus approximate the effect of suppression by treating these logits independently, providing a practical and interpretable proxy for class-sensitivity.

To ensure this computed factor is well-defined, we restrict the analysis to test samples $\{\mathbf{x}^{(n)}\}$ from class $i$ for which the following conditions hold:

$$z_i^{(n)} > 0 \quad \text{and} \quad R_i^{(n)} > 0. \tag{14}$$

**Positive logit requirement** $(z_i^{(n)} > 0)$   This ensures the sample is initially classified in favor of class $i$. Since we aim to compute the scaling required to suppress a confident prediction, it only makes sense to include samples where the logit is already positive.

**Positive relevance requirement** $(R_i^{(n)} > 0)$   This avoids division by zero and ensures that the latent features are, on balance, contributing positively to the logit. Including samples with non-positive relevance would contradict the assumption that scaling down their contributions leads to suppression.

To compute $\alpha_{\text{crit}}$ using this method, we follow this procedure:

1. **Compute per-image thresholds.**
   For each qualifying sample $\mathbf{x}^{(n)}$, compute $z_i^{(n)}$ and $R_i^{(n)}$ and then calculate:

   $$\alpha_{\text{crit}}^{(n)} = \frac{z_i^{(n)}}{R_i^{(n)}}.$$

2. **Aggregate across images.**
   Collect the set $\{\alpha_{\text{crit}}^{(n)}\}$ for all qualifying images $n$ for class $i$.

3. **Calculate summary statistics.**
   Calculate summary statistics, e.g., the median $\tilde{\alpha} = \text{median}_n\big(\alpha_{\text{crit}}^{(n)}\big)$.

A.7.2   PROCEDURE: NUMERICAL METHOD

To obtain a more precise estimate of the critical suppression threshold free from the inaccuracies of the linear approximation, we compute $\alpha_{\text{crit}}$ numerically. This approach directly finds the root of the exact modified logit equation:

$$z_i'(\alpha) = \sum_j w_{ij} \cdot \max(0, \, 1 - \alpha \, |c_j|) \cdot x_j = 0. \tag{15}$$

We restrict the analysis to a subset of samples that meet specific criteria to ensure the results are well-defined and interpretable.

**Positive logit requirement** $(z_i^{(n)} > 0)$   This requirement is identical to that of the analytical method and is maintained for the same reason: the concept of suppressing a logit to zero is only meaningful for samples that are already classified in favor of the target class.

**Existence of a Zero-Crossing**   While this method does not involve division, we must ensure that increasing $\alpha$ leads to suppression. A sufficient condition is that the function $z_i'(\alpha)$ is positive at $\alpha = 0$ (covered by the first requirement) and becomes negative for some sufficiently large $\alpha$. This guarantees that a root exists. Samples where the logit does not cross zero are excluded.

To compute $\alpha_{\text{crit}}$ numerically, we follow this procedure:

1. **Find the per-image root.**
   For each qualifying sample $\mathbf{x}^{(n)}$, we solve the equation $z_i'(\alpha^{(n)}) = 0$ for $\alpha^{(n)}$ using a simple and robust bracketing method. Specifically, we evaluate $z_i'(\alpha)$ over a discrete range of $\alpha$ values to find the interval where the function's sign changes from positive to negative. The precise root within this interval is then estimated using linear interpolation for higher precision.

2. **Aggregate across images.**
   Collect the set $\{\alpha_{\text{crit}}^{(n)}\}$ for all images $n$ of class $i$ for which a root was found.

3. **Calculate summary statistics.**
   Calculate relevant summary statistics from the collected set, e.g., the median $\tilde{\alpha}_{\text{crit}} = \text{median}_n\big(\alpha_{\text{crit}}^{(n)}\big)$.

This numerical procedure yields the true critical suppression threshold for each sample by avoiding the limitations of the linear approximation.

### A.7.3  Validation of the Analytical Approximation

The derivation of the analytical $\alpha_{\text{crit}}$ relies on a linear approximation that is valid when $1 - \alpha \cdot |c_j| > 0$ for all components $j$. To rigorously assess how frequently this condition holds, we calculated the full distribution of the term $1 - \alpha_{\text{crit}} \cdot |c_j|$. Specifically, for each per-sample numerical $\alpha_{\text{crit}}$, we calculated its interaction with every component of the corresponding $|c_j|$ vector, yielding the distribution of all pairwise suppression terms.

Figure 7 shows a clear architectural divergence. For the ResNet-18, the distribution is concentrated above zero, indicating that the linear approximation condition holds for the majority of samples. For the Vision Transformer, however, a significant mass of the distribution is in the negative region. This confirms that the linear approximation is a reasonable

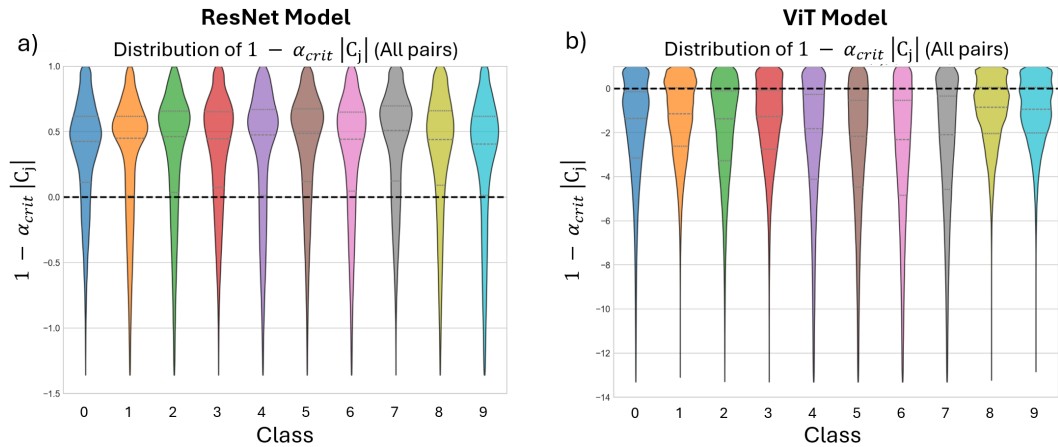

Figure 7: Validation of the linear approximation used to derive the analytical $\alpha_{\text{crit}}$. The plots show the distribution of the suppression term $1 - \alpha_{\text{crit}} \cdot |c_j|$, where negative values indicate the approximation is invalid. **(a)** For the **ResNet-18 model**, the distribution is concentrated above zero, confirming the approximation holds for the majority of samples. **(b)** For the **Vision Transformer**, a significant mass of the distribution is in the negative region, demonstrating that the linear approximation is invalid for the majority of samples.

approximation for the ResNet-18 but a poor one for the ViT. Consequently, we expect the analytical estimate of $\alpha_{\text{crit}}$ to represent a reliable lower bound for ResNet-18 but a significantly more conservative lower bound for the ViT. This is consistent with the results presented in the main text for ResNet-18 (Figure 5) and in the Appendix for the ViT (Figure 14)

## A.8 ROME METHOD FOR CLASS SUPPRESSION

To provide a baseline for permanent weight-editing approaches, we implemented a ROME-inspired rank-one update adapted for vision models. The original ROME method (28) was developed for factual editing in large language models by modifying mid-layer feed-forward networks. Our adaptation applies the same rank-one update principle to the final classification layer of a ResNet-18 for a class-suppression task, enabling a direct comparison to the final-layer edits performed by SALVE.

**Principle.** ROME treats a feed-forward layer as a key-value memory. For our use case, the key is the activation vector from the penultimate layer for a representative sample of the target class, and the value is a modified logit vector that strongly suppresses that class. Let $W \in \mathbb{R}^{C \times M}$ denote the weight matrix of the final fully connected layer, $k \in \mathbb{R}^M$ the key activation, and $v^* \in \mathbb{R}^C$ the desired output logits. The rank-one update is:

$$\Delta = \frac{(v^* - Wk)k^T}{\|k\|_2^2}, \qquad W' = W + \Delta. \tag{16}$$

For suppression, $v^*$ is defined by taking the original logits and setting the target class logit to a large negative value (e.g., $-10.0$). This update primarily affects the row corresponding to the target class, leaving other classes largely unchanged.

**Implementation Workflow.**

1. Extract the penultimate-layer activation for a representative sample of the target class.

2. Define the desired logit vector $v^*$ by strongly reducing the target class logit.

3. Compute the rank-one update $\Delta$ and apply it to the classification layer weights.

### A.8.1 PSEUDOCODE FOR ROME EDITING

```
function Apply-ROME-Edit(model, sample_image, target_class):
    # 1) Get activation from penultimate layer (key)
    key = get_penultimate_activation(model, sample_image)

    # 2) Define desired logits (value) with target class suppressed
    target_logits = model(sample_image)
    target_logits[target_class] = -10.0

    # 3) Compute rank-one update and apply to weights
    delta = outer_product(target_logits - (weights @ key), key) / dot(key, key)
    model.fc.weight += delta
```

### A.8.2 BENCHMARKING ROME FOR CLASS SUPPRESSION

To contextualize the performance of our feature-based weight modulation method (SALVE), we benchmarked it against ROME on the same class suppression task described in Section 4.2. As shown in Figure 8, both methods achieve similar outcomes: the target class accuracy is reduced to near zero while other classes remain largely unaffected. This demonstrates that SALVE is competitive with established weight-editing techniques for global interventions, while offering additional benefits of mechanistic interpretability and fine-grained feature control. Unlike ROME, which relies on a single-sample key and does not leverage latent feature structure, SALVE supports systematic edits across multiple concepts and enables per-sample sensitivity metrics such as $\alpha_{\text{crit}}$.

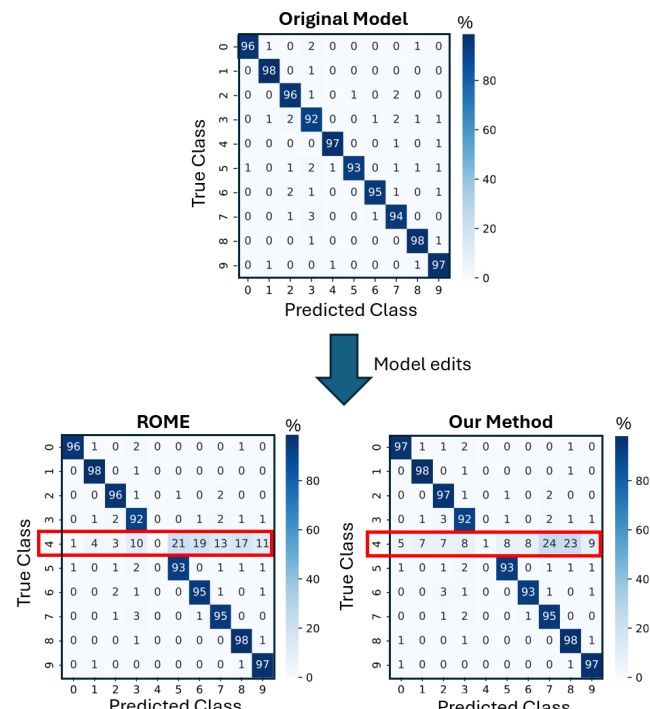

Figure 8: Comparison of class suppression performance. Confusion matrices showing predictions on the test set before editing (top) and after editing (bottom) with ROME and SALVE. Both methods successfully ablate the target class prediction (here shown for class 4).

## A.9  SAE-Based Activation Steering for Class Suppression

In addition to permanent weight modulation (Section 3.1), we introduce an alternative intervention method that leverages the same sparse autoencoder (SAE) representation to steer activations at inference time.

**Principle.**  Let $h \in \mathbb{R}^M$ denote the activation vector at a semantically rich layer $L$ (e.g., the global average pooling layer in ResNet-18). The SAE trained on $L$ provides a decoder matrix $D \in \mathbb{R}^{M \times d}$, where each column $D_{[:,l]}$ represents the direction of latent feature $l$ in the original activation space. To steer the model, we construct a vector

$$v = \sum_{l \in \mathcal{S}} \beta_l \, D_{[:,l]}, \tag{17}$$

where $\mathcal{S}$ is the set of selected latent features and $\beta_l$ are user-defined coefficients controlling the strength of the intervention. The edited activation becomes

$$\tilde{h} = h + v. \tag{18}$$

**Class-Targeted Suppression.**  To suppress a specific class $k$, we identify its dominant latent feature(s) using the class-conditional means of SAE activations, $\mu_k$ (as illustrated in Figure 1).

To reduce the influence of latent feature $l$, we apply a sign-aware coefficient:

$$\beta_l = -\beta \cdot \mathrm{sign}(\mu_{k,l}), \quad \beta \geq 0. \tag{19}$$

Steering is applied via a forward hook on layer $L$ during inference. After evaluation, the hook is removed, restoring the original model.

**Pseudocode for SAE Steering**

```
function Apply-SAE-Steering(model, sae, target_layer, target_class, beta):
    1) Identify the dominant latent feature for the target class
       (based on class-conditional mean activations)

    2) Determine the steering direction and strength

    3) Construct the steering vector from SAE decoder columns

    4) Temporarily apply the steering vector to the target layer
       (via a forward hook during inference)

    5) Run inference/evaluation to measure the effect of steering

    6) Remove the hook to restore the original model
```

### A.9.1 Benchmarking SAE-Based Activation Steering for Class Suppression

To contextualize the performance of our feature-based weight-editing method, we benchmarked an SAE-based Activation Steering approach on the same class suppression task described in Section 4.2.

As shown in Figure 9, both methods achieve highly comparable outcomes. The confusion matrices illustrate that Activation Steering effectively reduces the accuracy of the target class to near zero while leaving other classes largely unaffected (Figure 9b). To examine sensitivity to intervention strength, controlled by the parameter $\beta$, we evaluate performance across a range of $\beta$ values (Figure 9c). We further extend this analysis by computing suppression curves for the dominant feature of each class (Figure 9d), which exhibit trends similar to those observed for our method (cf. Figure 5a).

While these results are obtained on the relatively simple Imagenette benchmark, they indicate that SALVE is competitive with established inference-time steering techniques for global interventions, while additionally offering permanent edits and no overhead during inference. A further advantage of SALVE over steering lies in its ability to support per-sample diagnostics.

Standard linear steering methods typically apply a fixed additive offset along a concept direction, yielding interventions that are largely sample-independent. While effective for broad, class-level control, such approaches do not reveal how a particular input's activation pattern interacts with the concept.

Because SALVE's weight modulation interacts multiplicatively with the original activation pattern, the effect of an intervention depends on each sample's feature composition. This enables quantitative metrics such as $\alpha_{\mathrm{crit}}$, which reveal how strongly individual inputs rely on specific latent concepts—providing a principled mechanism for identifying brittle representations or samples vulnerable to adversarial perturbations.

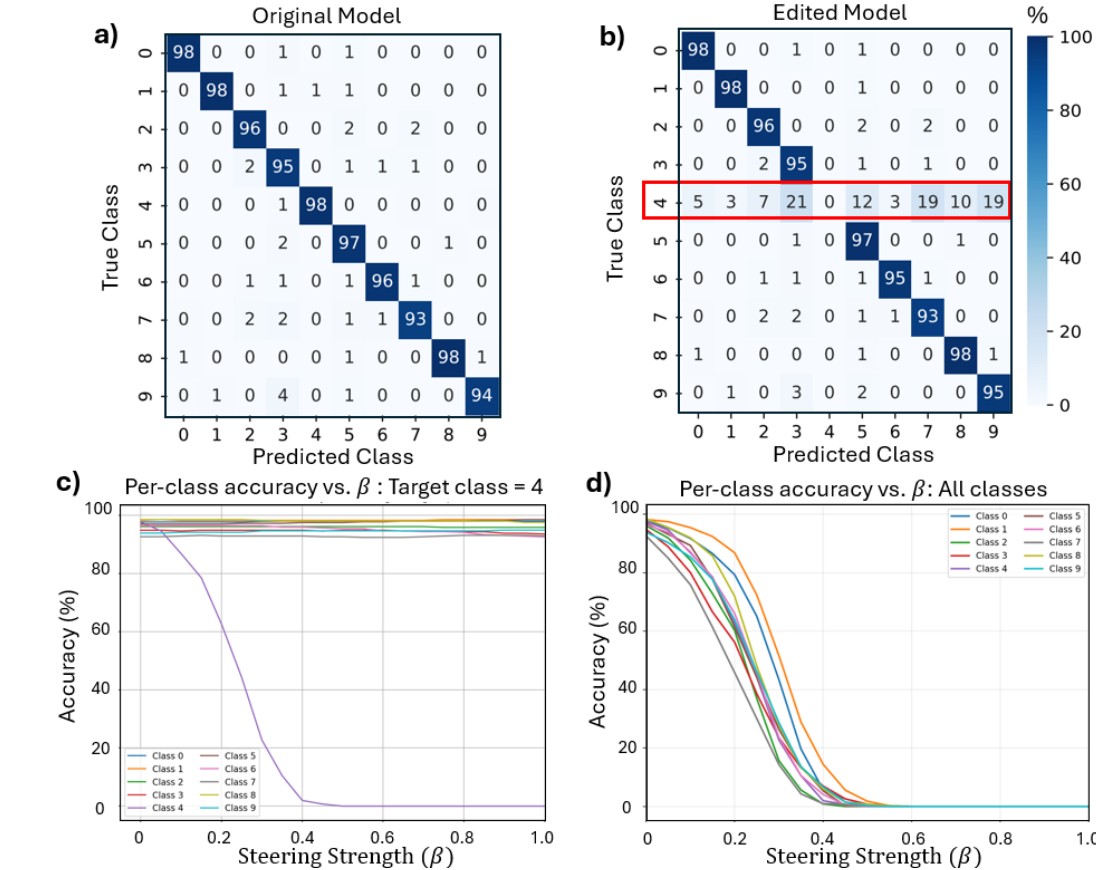

Figure 9: Comparison of class suppression performance. Confusion matrices showing the model's predictions on the test set before editing (a), and after editing (b) with SAE-based steering vectors. Both methods successfully ablate the target class prediction (here shown for class 4). (c): Suppression sensitivity to steering strength $\beta$ for class 4. (d): Suppression sensitivity to steering strength $\beta$ across all classes.

## A.10   VALIDATION ON CIFAR-100

To assess whether SALVE generalizes beyond small-scale datasets, we extend our analysis to CIFAR-100, which introduces substantially higher class diversity and therefore increases the difficulty of latent feature discovery and weight-space edits. We retrain the ResNet-18 classifier on CIFAR-100 using the same training parameters as for the Imagenette experiments. The resulting top-1 accuracy stabilizes at approximately 80%, consistent with previously reported results for this architecture. This lower baseline accuracy reflects the increased complexity of the dataset and provides a more challenging testbed for evaluating SALVE.

As a baseline, using the same SAE architecture and hyperparameters as in our Imagenette experiments (without any additional hyperparameter tuning for CIFAR-100), we find that the latent representation remains structured and relatively sparse (Figure 10a), though with noticeably more cross-class overlap than for Imagenette. This is expected: with 100 classes and richer visual variability, a simple linear $\ell_1$ SAE yields weaker feature separation and produces fewer strongly class-dominant latents. Replicating our latent-feature analysis, several latents exhibit activation patterns shared across semantically related classes. For example, Latent 4 is strongly activated for Class 2 (Baby) but also for Classes 11 (Boy) and 35 (Girl). Other latents remain more class-dominant, such as Latent 19, which is primarily associated with Class 1 (Aquarium Fish). As in our Imagenette study, we use this dominant latent feature to perform targeted model edits.

To enable a direct comparison between post-hoc weight editing and runtime steering, we perform the same suppression experiment using both SALVE (Figure 10b) and activation steering (Figure 10c). In both cases, we use the same SAE-derived latent feature, allowing us to assess how each intervention behaves when grounded in the same feature basis.

Figure 10b shows that suppressing this feature reduces the accuracy of the corresponding class in a manner consistent with our earlier experiments. Here, the accuracy for the target class is shown in red, whereas the other classes are represented in grey. Off-target effects remain limited for moderate values of $\alpha$, while higher values induce more drift due to the weaker feature separation obtained by this SAE configuration for the CIFAR-100 dataset. Figure 10c exhibits a similar trend for activation steering, where increasing the steering strength $\beta$ reduces the target-class accuracy in a comparable manner.

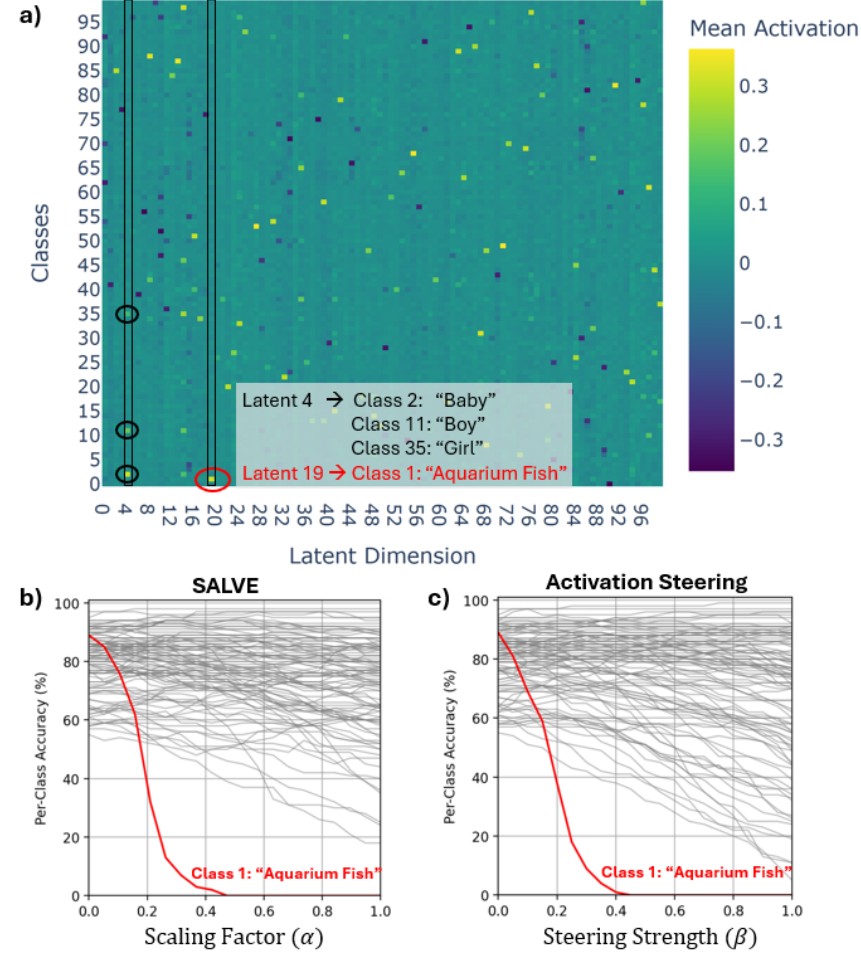

Figure 10: **a)** Validation of SAE-discovered features for the ResNet-18 model on CIFAR-100. Average latent activations across classes reveal meaningful structure, though with substantially more cross-class overlap than in Imagenette (e.g., Latent 4 activates across multiple human-related classes, whereas Latent 19 is primarily associated with the Aquarium Fish class). **b–c)** Per-class accuracy when suppressing the dominant latent feature associated with Class 1 (Aquarium Fish), using the same SAE-derived feature basis for both SALVE (b) and Activation Steering (c). The target class is shown in red and all other classes in grey.

Overall performance is reduced relative to Imagenette, reflecting the increased difficulty of obtaining clean feature separation in a 100-class setting with richer underlying visual variability. Importantly, this reduction in edit performance should not be interpreted as a limitation of SALVE itself, as activation steering applied to the same SAE basis exhibits nearly

identical suppression patterns. This indicates that the dominant factor governing edit quality on CIFAR-100 is the weaker disentanglement of the underlying SAE representation, rather than the specific editing mechanism. Both approaches perform similarly when grounded in the same latent feature, further supporting that improvements in the autoencoder architecture, and not the editing method, are the primary pathway toward better controllability on larger, more complex datasets.

Scaling SALVE to larger datasets or more complex architectures will likely benefit from stronger SAE variants (e.g., JumpReLU, gated, or top-$k$), which may improve reconstruction fidelity and feature disentanglement. These scaling experiments represent an important extension of our work, as discussed in the Limitations section. Importantly, SALVE is SAE-agnostic, and these alternative autoencoder variants can be integrated directly without modifying the control mechanism or the per-sample diagnostic components of the pipeline.

## A.11   Vision Transformer Results

This section provides the results for the complementary validation analysis performed on the Vision Transformer (ViT-B/16) model.

### A.11.1   Latent Features Encode Semantic Concepts

To confirm the generality of our feature discovery method, we first replicate the analysis of the learned feature basis on the ViT. As illustrated in Figure 11a, the SAE successfully learns a sparse and class-specific latent basis, analogous to the one found for ResNet-18. We

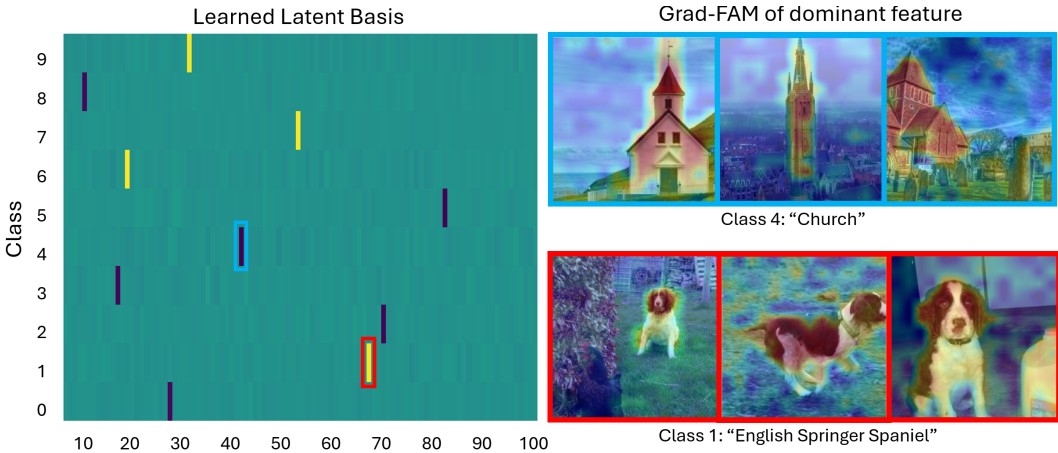

Figure 11: Validation of discovered features for the ViT-B/16 model. (a) Average latent feature activations across classes, highlighting a strong class-specific structure. (b) Grad-FAM visualizations grounding the dominant features for Class 4 ("Church") and Class 1 ("English springer") in representative images.

then validate the semantic meaning of these discovered features using Grad-FAM to visualize their grounding in specific input images. Figure 11b shows examples for two class-dominant features, where the saliency maps correctly highlight the class-relevant objects in the images (the church building and the English springer spaniel). This provides strong evidence that the SAE uncovers semantically meaningful and spatially localized concepts for the Vision Transformer, just as it does for the CNN.

### A.11.2   Controlling Class Predictions via Feature Editing

To validate the architectural robustness of our control method, we replicate the class suppression experiments on the Vision Transformer. The results, shown in Figure 12, confirm that our feature-guided intervention is effective also on the transformer-based model.

Specifically, suppressing the dominant feature for "Church" (Class 4) reduces its accuracy on the test set to near zero. Critically, the performance on other classes remains largely unaffected, as shown in the confusion matrix. This provides strong evidence that the features learned by the ViT also act as modular switches, allowing for precise, targeted interventions with minimal off-target effects. We observe similar successful suppression across the other classes in the dataset.

### A.11.3   Quantifying Intervention Sensitivity

To investigate the sensitivity of the interventions also for the Vision Transformer, we replicate the suppression analysis from the main text.

First, we analyze the characteristic suppression curve for a single class. As shown in Figure 13, systematically increasing the intervention strength $\alpha$ results in a stable accuracy that drops

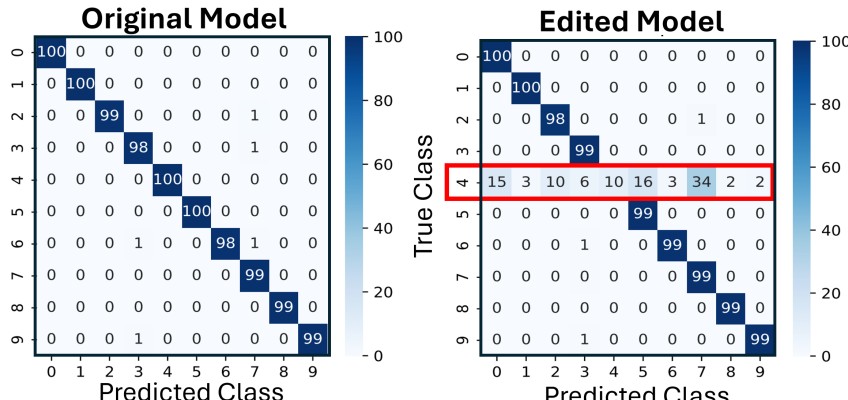

Figure 12: Quantitative validation of feature-guided control in the ViT-B/16 model. The confusion matrices show performance on the test set before (left) and after (right) suppressing the dominant feature for the "Church" class. The intervention effectively ablates the target class while preserving accuracy on the other classes.

sharply past a critical threshold, while the model's predictions are reallocated to other classes. This shows that the intervention exhibits the same targeted behavior for the ViT as it did for the ResNet-18.

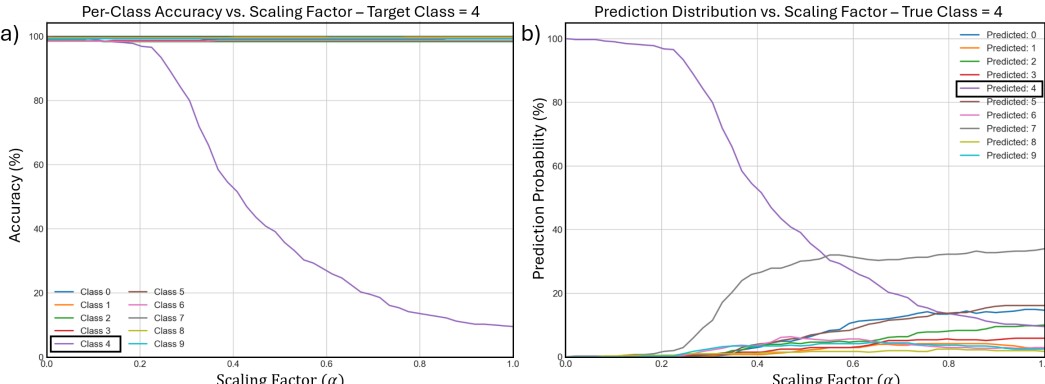

Figure 13: Suppression sensitivity for a single class (Class 4, "Church") in the ViT model. **(a)** Per-class accuracy as a function of the scaling factor $\alpha$. **(b)** Distribution of predictions for images of the target class, showing how confidence is reallocated as the feature is suppressed.

Next, we extend this analysis to all classes and compute the analytical, numerical, and empirical estimates for $\alpha_{\mathrm{crit}}$. Figure 14a shows the full set of per-class suppression curves, which exhibit a wider spread in thresholds compared to the ResNet-18, though all classes could be suppressed to near-zero accuracy. Figure 14b presents the comparison of the $\alpha_{\mathrm{crit}}$ estimates. This plot confirms two key findings also discussed in the main text.

First, the analytical estimate of $\alpha_{\mathrm{crit}}$ represents a much more conservative lower bound. Second, we observe a larger discrepancy between the numerically calculated $\alpha_{\mathrm{crit}}$ (representing total evidence loss) and the empirical $\alpha_{50\%}$ (representing a prediction flip) for the ViT. We hypothesize that this discrepancy can be attributed to the ViT's architectural properties. Recent research has shown that ViTs learn a "curved" and non-linear representation space (20). This effect, combined with the ViT's tendency to produce less confident, more distributed logits (30), means a smaller intervention is required to be surpassed by a competitor. This widens the gap between the point of prediction failure ($\alpha_{50\%}$) and total evidence loss ($\alpha_{\mathrm{crit}}$), an effect less pronounced in the more linear representations of the ResNet model.

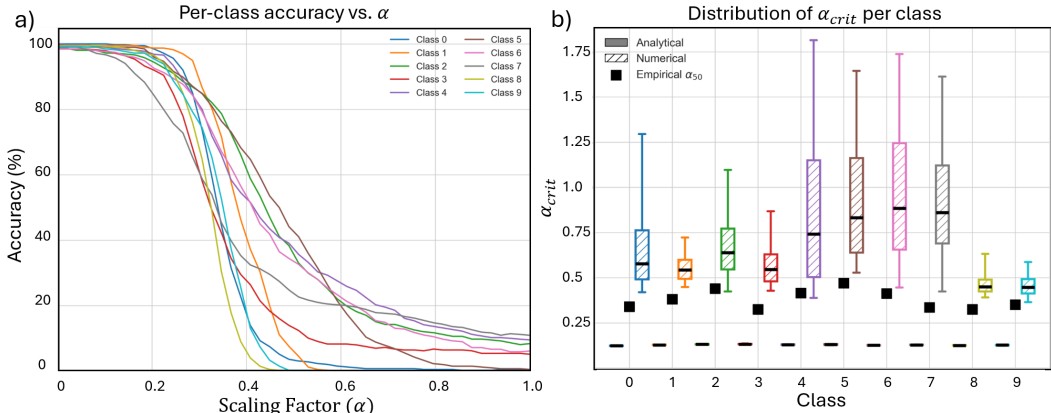

Figure 14: Full class-level sensitivity analysis for the ViT model. **(a)** Per-class accuracy vs. intervention strength $\alpha$ for all classes. **(b)** Comparison of analytical (filled), numerical (hatched), and empirical (square marker) $\alpha_{\text{crit}}$ estimates, illustrating the inaccuracy of the linear approximation for the ViT.

Another factor that can contribute to the observed wider spread in $\alpha_{\text{crit}}$ thresholds and the increased difficulty in driving some classes to near-zero accuracy, is that of feature entanglement. When we suppress a single dominant latent feature associated with a class, other entangled features might still contribute to the same class's logit. This makes it intrinsically harder to drive the total feature contribution for that class's logit completely to zero. This effect would result in larger required $\alpha_{\text{crit}}$ values and can explain instances where suppression curves, as seen in Figure 14a, level off above zero accuracy even at high $\alpha$. The combined characteristics of the model backbone's feature representation and the SAE's ability to extract a sparse decomposition can thus critically influence intervention efficacy. Further discussion on these intertwined effects is provided in Section A.12.2.

## A.12 ADDITIONAL EXPERIMENTS ON CLASS-ROBUSTNESS

This section presents supplementary experiments assessing the sensitivity of the class-robustness results.

### A.12.1 ROBUSTNESS ACROSS AUTOENCODER INITIALIZATIONS

We trained multiple autoencoders ($n = 10$) with identical architectures and loss parameters, but different random seeds, weight initializations, and data shuffling during training. This led each autoencoder to learn a distinct latent basis, as illustrated in Figure 15a.

For each learned representation, we repeated the experiments from Section 4.4, computing accuracy–vs–$\alpha$ curves (Figure 15b) and analyzed how model confidence redistributed across the classes (Figure 15c). Shaded regions in these plots indicate the standard deviations across the different realizations.

Despite learning distinct latent bases, the resulting suppression curves were nearly identical, and the same subset of classes remained most resistant to suppression. This observation aligns with findings on the stability of features learned by sparse autoencoders trained on the same data (36), suggesting that the heterogeneity in class robustness is not an artifact of a particular latent decomposition due to random initialization.

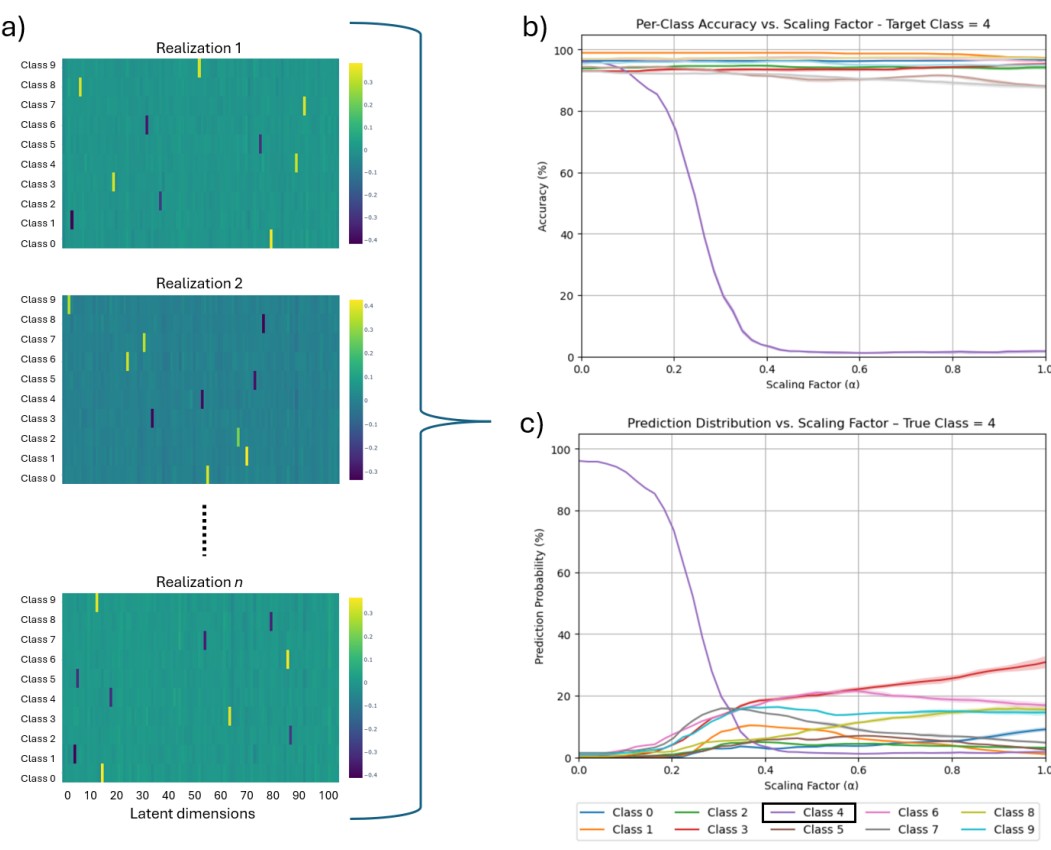

Figure 15: a) Learned latent bases across the $n = 10$ realizations. b) Per-class prediction accuracy as a function of the scaling factor $\alpha$, when suppressing the dominant feature associated with class 4. c) Distribution of predictions among the different classes, where the "true" class is 4, showing how the models confidence becomes re-distributed among the other classes. The shaded region in these plots represent the standard deviation across $n = 10$ different realizations.

This indicates that for a fixed SAE training objective, the observed differences in class sensitivity stem mainly from the base model's intrinsic feature representations. However, the training objective itself, particularly the sparsity coefficient $\lambda_1$, fundamentally shapes the feature decomposition. Therefore, the observed heterogeneity is a product of both the backbone's intrinsic structure and the specific features learned by the SAE. While a detailed analysis of the impact of the SAE training objective is a key topic for future work, the role of the backbone model is addressed further in the next section.

### A.12.2  ROBUSTNESS ACROSS BACKBONE MODELS

To examine whether the backbone's learned feature space contributes to the observed class heterogeneity, we fine-tune several ResNet-18 variants on Imagenette with different random initializations and training parameters.

Prior work has reported that minor changes in training can alter class robustness profiles (2; 26). Consistent with these findings, we observe that the "hard-to-suppress" classes varies across backbones (Figure 16), suggesting that the base network's feature space is the dominant factor determining class heterogeneity.

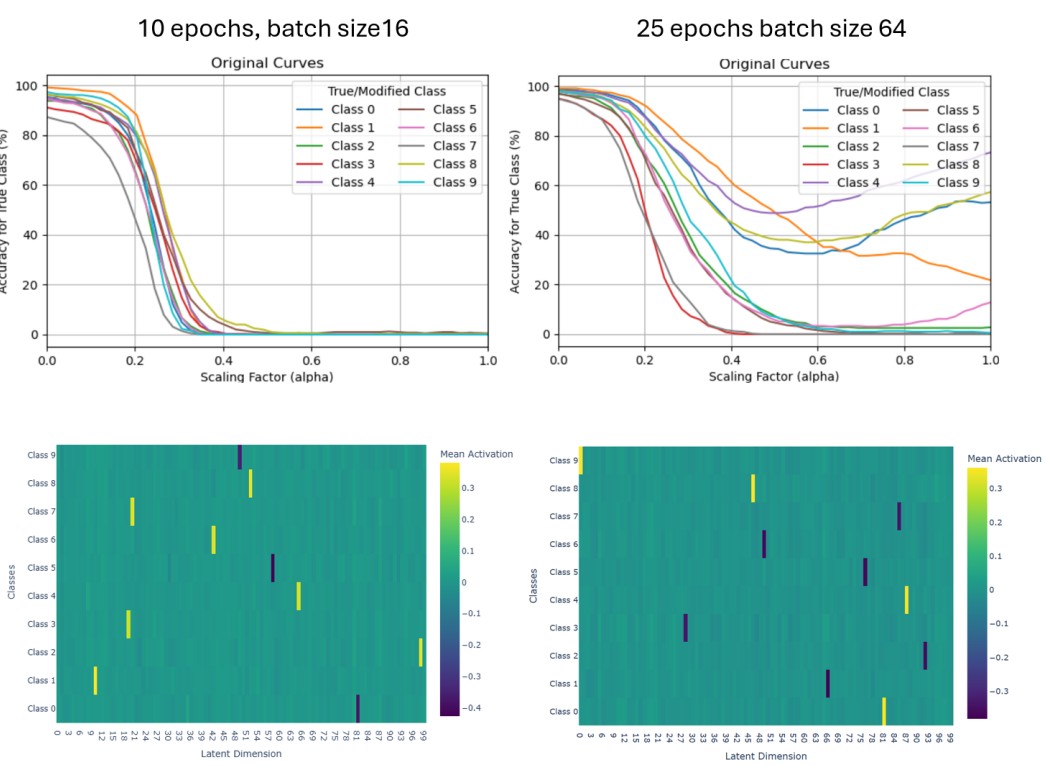

Figure 16: Accuracy vs. scaling factor $\alpha$ for ResNet-18 backbones trained with different initialization and fine-tuning parameters.

In our experiments, we find that the optimizer's batch size during fine-tuning of the backbone is a key parameter that strongly affects class suppression sensitivity. When using small batches (e.g. 8–16), all classes can be driven to near-zero accuracy with moderate $\alpha$, and the suppression curves remain tightly aligned in both low- and high-$\alpha$ regimes. In contrast, large batches (e.g. 64–128) substantially increases the heterogeneity of suppression, where $\alpha_{\text{crit}}$ grows larger for many classes and only a subset of classes could be fully suppressed.

Some experiments also exhibit more complex suppression curves, where certain classes are only partially suppressed even under high attenuation, and a few show non-monotonic behavior, where accuracy initially drop, then partially recover at higher $\alpha$. These suppression

behaviors resemble the promotion-suppression effects observed in previous work on adversarial perturbations, where certain features are either suppressed or promoted, leading to changes in model predictions (48).

We hypothesize that these effects may rise from the well-known impact of batch size on the geometry of the loss landscape (18). Large-batch training tends to converge to sharper minima, which are known to produce more entangled internal representations. These entangled features are naturally harder to isolate and suppress with our targeted method. In contrast, the stochasticity of small-batch training acts as an implicit regularizer, guiding the model to flatter minima associated with more disentangled and modular features (15). In these solutions, concepts are more cleanly separated, allowing our interventions to suppress class predictions more effectively.

Beyond backbone training, our experiments also indicate that the parameters used for training the Sparse Autoencoder (SAE) significantly influence the learned latent feature representations and, consequently, their downstream controllability. Specifically, the $\ell_1$ regularization loss plays a critical role: too low $\ell_1$ values can lead to a less sparse, entangled latent space where multiple features activate for a single class, making targeted suppression difficult. Conversely, overly high $\ell_1$ values may result in some class-specific features failing to activate at all. Achieving an effective balance between reconstruction accuracy and $\ell_1$ sparsity is crucial, and the optimal parameter values depends on the specific backbone architecture and its training dynamics, necessitating tailored values for models like both ResNet and ViT.

Together, these findings highlight a critical link between training dynamics, architectural choices, and downstream controllability. They suggest that co-designing training regimes and architectures to favor flatter, more modular solutions may be a key strategy for developing models that are not only performant but also inherently more interpretable and editable. While our initial analysis provides evidence for this connection, a more comprehensive exploration of batch size, alternative SAE variants, and other training parameters on feature modularity and intervention efficacy represents a crucial direction for future work.

