# OpenReview forum: "SALVE: Sparse Autoencoder-Latent Vector Editing for Mechanistic Control of Neural Networks"
_ICLR.cc/2026/Conference — Submitted to ICLR 2026_

### Official Review · Reviewer_XJkk · 2025-10-15

**Soundness:** 2
**Presentation:** 3
**Contribution:** 1
**Rating:** 2
**Confidence:** 4

**Summary:**

The current work proposes a three-step process meant to introduce a permanent change in the weights of a given model aiming to control its behavior:

**Step 1 (Discover):** First, a Sparse Autoencoder (SAE) is trained to decompose activations. According to the description provided in Appendix 4, it is a set of two linear layers (an encoder and a decoder), featuring no activation functions,  trained to minimize the reconstruction loss with an L1 penalty applied on the latent SAE features.

**Step 2 (Validate):** The authors apply Grad-CAM on activations from intermediate representation provided by the SAE to determine whether or not a certain feature is semantically relevant for the given task.

**Step 3 (Control):** In the final step the weights of the original model are adjusted by means of adding or subtracting the weights (associated to the selected SAE features) of the SAE decoder layer from the original model’s weights.

Experiments are performed on a Resnet-18 and a ViT-B/16 using the Imagenette benchmark showing that the method is able to identify features that are relevant for certain classes and to steer the decisions of the model.

**Strengths:**

- The paper is well written, clearly structured and presented.
- The evaluation is performed both on a CNN and a Transformer model.
- The results show that the method is able to steer the decisions of the models under the circumstances of the considered setup.

**Weaknesses:**

**Step 1 (Discover):** It is unclear why the authors have chosen to employ an autoencoder that differs significantly both in terms of structure and in terms of the training objective from the current state-of-the-art SAEs \[1, 2\]. The chosen design and training objective is not motivated in the paper and these decisions do not seem to properly model the goals of an SAE.

**Step 2 (Validate):** The proposed validation step requires human intervention which is not feasible in the context of large models and it is not an improvement over current automatic feature selection methods \[3, 4\].

**Step 3 (Control):** It is unclear why the standard approach of hooking an SAE module to an existing transformer is any less permanent than a weight change. The authors could argue that it is more computationally efficient to change the weights directly rather than passing the activations through an SAE, but not that it is more permanent. Furthermore, the procedure proposed in Eq. 1 seems to be similar to what SAEs regularly do. In a regular setup, increasing or decreasing an SAE latent results in adding or subtracting a decoder weight from the initial activations. Similarly, in Eq. 1, the procedure simply adds or subtracts a decoder weight. While changing the weights rather than the activations is more efficient, it comes with the downside that the addition/subtration proposed in Eq. 1 is always performed with a fixed coefficient, while in a regular setup that coefficient is adjusted for each sample based on the latent SAE representation.

**Evaluation:** The current evaluation procedure is (i) restricted to two small models, (ii) restricted to a classification setup, (iii) restricted to a small benchmark and (iv) lacking a proper comparison with respect to existing approaches. While the current work cites a sum of methods aimed at model control, it does not provide a proper comparison between the proposed approach and the existing ones. In order to prove the efficiency of the proposed approach one or more of the following could have been employed: (i) showing that the approach can steer the prediction of a model towards a gender-neutral one without affecting hair color prediction on CelebA \[5\], (ii) showing that the proposed approach can effectively steer language models \[6\] towards exhibiting desired behaviors, (iii) showing that the proposed approach can effectively steer diffusion models \[7\].

\[1\] Rajamanoharan et al., “Jumping Ahead: Improving Reconstruction Fidelity with JumpReLU Sparse Autoencoders”
\[2\] https://transformer-circuits.pub/2025/january-update/index.html
\[3\] https://transformer-circuits.pub/2024/scaling-monosemanticity/
\[4\] Cywinski et al., “SAeUron: Interpretable Concept Unlearning in Diffusion Models with Sparse Autoencoders”
\[5\] Gerych et al., “BendVLM: Test-Time Debiasing of Vision-Language Embeddings”
\[6\] Kim et al., “Interpretability Beyond Feature Attribution: Quantitative Testing with Concept Activation Vectors (TCAV)”
\[7\] Surkov and Wendler et al., “One-Step is Enough: Sparse Autoencoders for Text-to-Image Diffusion Models”

**Questions:**

[Step 1] Why was the current SAE implementation chosen and how does it compare to existing approaches?

[Step 2] Can the authors provide a comparison between their selection mechanism and existing ones, highlighting the improvements brought in this regard?

[Step 3] Can the authors provide an in depth analysis between their weight space approach and the existing approaches to using SAE for steering?

---

> ### Author Response · Authors · 2025-11-19
>
> We appreciate your careful reading and constructive suggestions. We respond point by point below.
>
> ### Step 1 (Discover)
>
> Our current scope isolates SALVE’s core mechanisms in a controlled setting. Our vision models have low latent dimensionality, making sparse feature extraction with good reconstruction fidelity tractable with a basic SAE with L1 sparsity. Scaling to larger datasets or more complex architectures will likely benefit from SAE variants more widely used in large-scale LLM interpretability (e.g., JumpReLU, gated, top-k). We note that SALVE is SAE-agnostic, and these variants can be integrated without modifying the control or diagnostic components.
> We have expanded the “Related work” and “Limitations and Future Work” sections to clarify this.
>
>
> ### Step 2 (Validate)
>
> Our validation focuses on interpretability and grounding rather than automated selection. Many weight-editing methods, including ROME and MEMIT, lack a semantic validation mechanism. While activation steering can in principle use the same interpretable SAE basis, it faces the same challenge of manual feature validation.
> We agree that automated selection is important and represents a broader challenge not unique to our method. As such, this remains an important topic for the wider research community.
>
>
> ### Step 3 (Control)
>
> **Why “permanent”:**
> Hook-based SAE steering and other activation-time interventions are runtime-dependent, requiring hooks and auxiliary modules during inference. While activation-time methods demonstrate flexible control, they operate as overlays rather than persistent model states. In contrast, SALVE modifies weights directly, producing a single, auditable artifact, a distinction particularly relevant for compliance-sensitive or deployment-constrained settings.
> The revised “Related Work” and Table 1 is expanded to include recent SAE-steering approaches, as well as alternative model editing methods, and we have clarified the value proposition of our proposed method for weight editing.
>
> **Comparison to SAE Steering:**
> The reviewer notes that SAE steering is adjusted for each sample. This is accurate for encoder-gated variants, where steering is scaled by the latent activation $\phi_l(x)$. However, many methods, including those used in recent VLM and LLM interventions, apply fixed offsets, making them sample-independent in practice. SALVE differs by applying a fixed modulation to weights, but retaining sample-dependency through the activations $x$.
>
> Eq. (1) applies a multiplicative mask:
>
> $\tilde W_{ij} = W_{ij} \cdot \text{max}(0, 1 \pm \alpha c_j)$ along a feature direction. This is not equivalent to adding a decoder column to activations. With a fixed $\alpha$ per edit, the net effect remains input dependent because logits combine $\tilde W_{ij}$ with per-sample activations $x_j$. This enables per-sample diagnostics such as $\alpha_{\text{crit}}$, representing a capability not natively supported by standard steering approaches.
> We clarify this key distinction in the revision.
>
> ---
> ### Evaluation
>
> The “Related Work” section and Table 1 is expanded to include recent SAE-steering approaches and alternative model-editing or concept-based intervention methods. While comprehensive benchmarking across all methods is out of scope, we compared SALVE to two representative classes: ROME for permanent weight edits and SAE-based activation steering for inference-time control.
>
> **Benchmarking**
>
> We add new experiments comparing SALVE to activation steering using the same SAE-extracted features (revised Appendix A.9). Results show similar class suppression performance, while SALVE offers additional diagnostic capabilities.
>
> **Validation on CIFAR-100**
>
> To assess generalization beyond small-scale datasets, we added new experiments on CIFAR-100 (revised Appendix A.10). Although feature disentanglement is more challenging, given the higher class diversity, SALVE still discovers meaningful directions and supports effective edits.
>
> We agree that broader evaluation across datasets, architectures, and modalities is important. Extending SALVE to tasks such as CelebA attribution steering, LLM behavior control, and diffusion models is valuable future work but requires substantial new pipelines. We have expanded the Related Work section in the revision and note in Limitations the need for further robustness benchmarks and cross-modality evaluation.
>
> ---
> ### Closing
>
> Our current scope isolates SALVE’s core mechanisms in a controlled setting. While evaluation is limited to relatively simple models and datasets, we argue this work provides a foundation for future extensions and benchmarking across architectures and modalities.
> Our main takeaway is that SALVE, even in its current form, achieves comparable performance to established steering and editing methods while introducing additional diagnostic capabilities.

---

### Official Review · Reviewer_nMGe · 2025-10-22

**Soundness:** 2
**Presentation:** 2
**Contribution:** 2
**Rating:** 2
**Confidence:** 4

**Summary:**

This paper proposes using a sparse autoencoder to discover interpretable concepts within a classification model, identifying class-related concepts. Based on these concepts, it edits the final layer to alter model outputs. The method is validated on both ResNet and ViT architectures using the Imagenette dataset.

**Strengths:**

1. The paper introduces an interpretable model editing method to directly adapt model behavior.
2. Intervention sensitivity analysis establishes a quantitative suppression threshold to calibrate model edits.

**Weaknesses:**

1. Using SAEs to identify concepts and highlight relevant image regions has been established, such as [1]. Similarly, suppressing model components to adapt model behavior has been explored in works like [2]. Therefore, the technical novelty of this work seems limited.

2. More extensive comparisons with recent model editing or concept-based intervention methods (e.g., [3]) are needed to better demonstrate potential advantages.

3. Experiments are conducted on the relatively small-scale Imagenette dataset, raising concerns about scalability to larger datasets like ImageNet.

[1] Sparse Autoencoders Reveal Selective Remapping of Visual Concepts during Adaptation, ICLR 2025

[2] SAeUron: Interpretable Concept Unlearning in Diffusion Models with Sparse Autoencoders, ICML 2025

[3] Decomposing and Editing Predictions by Modeling Model Computation, ICML 2024

**Questions:**

1. Why focus only on editing the final layer? Could exploring SAE concepts and editing in earlier layers enable adapting the model's behavior more effectively?

---

> ### Author Response · Authors · 2025-11-19
>
> We thank the reviewer for their constructive feedback and for recognizing the interpretability and diagnostic contributions of our approach.
>
> ### Novelty and Positioning
>
> SALVE builds on established components such as sparse autoencoders (SAEs), activation maximization, and weight-space interventions, but its novelty lies in integrating these elements into a unified, post-hoc pipeline for interpretable, permanent model control. This enables capabilities not achievable in isolation:
>
> - Feature discovery → validation → control occurs in a single latent space, grounding edits in semantically validated features.
> - Diagnostics: SALVE’s edits are permanent and sample-dependent, interacting multiplicatively with the model’s learned weights. This distinction enables fine-grained control and diagnostics via $\alpha_{\text{crit}}$, supporting per-sample robustness auditing, a capability not natively supported by alternative steering approaches.
>
> To our knowledge, SALVE is the first framework to connect these steps into a methodology for validated, feature-driven weight edits.
>
>
>
> ### Benchmarking
>
> We have added new experiments directly comparing SALVE to activation steering using the same SAE features. These results (included in the revised Appendix A.9) show comparable class-suppression performance, while SALVE offers additional capabilities.
>
> **Weight edits vs steering:**
> Hook-based SAE steering and other activation-time interventions are runtime-dependent, requiring auxiliary modules during inference. In contrast, SALVE modifies weights directly, producing a single, auditable artifact. This distinction is particularly relevant for compliance-sensitive or deployment-constrained settings.
>
> **Comparison to SAE Steering:**
> Many SAE steering methods, including those used in recent VLM and LLM interventions, apply fixed additive offsets, making them sample-independent in practice. SALVE differs by applying a fixed modulation to weights, but retaining sample dependency through the activations $x$.
>
> Eq. (1) applies a multiplicative mask $\tilde W_{ij} = W_{ij} \cdot \text{max}(0, 1 \pm \alpha c_j)$ along a feature direction. This is not equivalent to adding a decoder column to activations. Importantly, with a fixed $\alpha$ per edit, the net effect remains input dependent because logits combine $\tilde W_{ij}$ with per-sample activations $x_j$ (i.e., the edit interacts with each input’s feature composition). This enables per-sample diagnostics such as $\alpha_{\text{crit}}$, representing a capability not natively supported by standard steering approaches.
>
> These key distinctions are clarified in the revised “Related Work” and “Methods” sections.
>
>
>
> ### Evaluation Scope
>
> The “Related Work” section and Table 1 have been expanded to include recent SAE-steering approaches and alternative model-editing or concept-based intervention methods, including the suggested computation-path editing work.
> While comprehensive benchmarking across all these methods is out of this submission’s scope, we compare SALVE to two representative classes: ROME for permanent weight edits and SAE-based activation steering for inference-time control.
>
> To assess generalization beyond small-scale datasets, we added new experiments on CIFAR-100 (included in revised Appendix A.10). Although feature disentanglement is more challenging given the higher class diversity, SALVE still discovers meaningful directions and supports effective edits. Scaling to larger datasets or architectures will likely benefit from stronger SAE variants (JumpReLU, gated, top-k), which can be integrated without modifying SALVE’s control or diagnostic components.
> We clarify this in the revised Limitations and Future Work.
>
>
>
> ### Edit depth
>
> We apply edits at the final layer because this provides a clear mapping from SAE-decoded features to classifier decisions. Deeper-layer editing is a promising but nontrivial extension, as such layers involve convolutional filters (ResNet) or attention/MLP blocks (ViT), making the mapping between SAE features and weight edits more complex. We view this as a promising extension and have noted it as future work in the Limitations section.
>
> ---
>
> ### Closing
>
> Our current scope isolates SALVE’s core mechanisms in a controlled setting. While evaluation is limited to relatively simple models and datasets, we argue this work provides a foundation for future extensions and benchmarking across architectures and modalities. This is beyond the scope of a single paper but represents a promising direction we believe to be of interest to the broader research community.
>
> Our main takeaway is that SALVE, even in its current form, achieves comparable performance to established steering and editing methods while introducing additional diagnostic capabilities.
> We hope these clarifications and the additional experiments address the reviewer’s concerns regarding evaluation scope and SALVE’s broader value proposition relative to alternative approaches.

---

### Official Review · Reviewer_tRBT · 2025-10-30

**Soundness:** 2
**Presentation:** 3
**Contribution:** 2
**Rating:** 2
**Confidence:** 4

**Summary:**

The paper introduces SALVE (Sparse Autoencoder-Latent Vector Editing), a three-stage pipeline to discover, validate, and control model-native features. A linear SAE with an L1 penalty is trained on late-layer activations to learn a sparse basis; feature semantics are visualized via activation maximization and a proposed Grad-FAM saliency that targets latent features. The same SAE decoder provides directions for permanent weight-space edits that suppress or enhance specific features, and the paper defines $\alpha_\mathrm{crit}$ to quantify how strongly a class relies on a feature. Experiments target ResNet-18 and ViT-B/16 fine-tuned on Imagenette, with class-specific and cross-class interventions and small ablations against a ROME-style baseline.

**Strengths:**

* Originality: Clear, cohesive pipeline that connects unsupervised SAE features to weight-space edits, not only inference-time steering. The $\alpha_\mathrm{crit}$ metric offers a concrete knob to quantify reliance per sample, which is useful for diagnostics
* Quality: Solid derivation for the analytic approximation of $\alpha_\mathrm{crit}$ with a numerical check; careful discussion about when the linear approximation is reasonable for ResNet versus ViT
* Clarity: Method is easy to follow, with a simple linear SAE, explicit loss, and pseudocode for Grad-FAM and the numerical root finding
* Significance: If scaled and benchmarked broadly, the approach could offer a practical route to permanent, auditable edits tied to interpretable features, which is attractive for safety and compliance scenarios

**Weaknesses:**

* Only Imagenette is used and only two backbones are tested. There is no evaluation on harder datasets, no distribution shift stress tests, no adversarial or corruption benchmarks, and no human studies for interpretability quality. The ROME comparison is minimal and customized to the final layer rather than the standard internal-layer setting
* The discovery component is a standard linear SAE with L1 sparsity. Grad-FAM adapts Grad-CAM to a latent feature target, which is straightforward. The control step scales weights along decoder directions, which closely resembles linear readout manipulation or feature-direction ablations in the logit space. The contribution is mainly the packaging and the diagnostic $\alpha$ metric, not a fundamentally new mechanism
* Table 1 is helpful but misses several current concept-level and editing baselines in vision and language, such as more recent SAE-steering works and concept-bottleneck variants with post-hoc heads. The empirical section does not compare against activation-space steering on equal footing, INLP-style projections on the same backbone, or pruning-based continuous ablations that preserve accuracy.
* All edits appear to target the classifier head through penultimate activations; it is not shown how deeper-layer edits behave, how much performance drift occurs on unrelated classes in larger settings, or whether repeated edits compose cleanly. The paper acknowledges this as future work.

**Questions:**

* Can you expand Table 1 to include recent SAE-steering for ViTs and CLIP, editing methods beyond ROME and MEMIT, and causal feature interventions that operate in hidden layers, then compare empirically on shared metrics (e.g.: https://arxiv.org/abs/2504.08729, https://proceedings.mlr.press/v267/zaigrajew25a.html)?
* Your edit multiplies final-layer weights by a function of the decoder column for a feature. Please clarify whether SALVE always acts on the last layer and, if so, how this differs materially from directly editing a linear readout vector or from low-dimensional logit steering; ideally show deep-layer edits with the same decoder direction mapped backward.
* Please benchmark against activation-space steering with identical features and an equivalent compute budget, and report accuracy, off-target effects, and edit permanence. This will isolate the value of permanent weight edits versus temporary steering.
* Can you repeat the study on CIFAR-100 or ImageNet subsets, and include distribution shift tests and corruption robustness, reporting accuracy drops and off-target drift under increasing alpha? alpha should also be connected to adversarial susceptibility
* Your ROME adaptation targets the classifier head. Can you also evaluate a mid-layer ROME or MEMIT edit to match the original setting and show confusion matrices for both, plus your method, on the same task?
* Beyond visuals, can you measure concept-localization quality numerically, for example with Network Dissection-style overlaps, TCAV sensitivity for user-named concepts, or segmentation IoU where labels exist?

---

> ### Author Response · Authors · 2025-11-19
>
> We thank the reviewer for their constructive feedback and for recognizing the clarity, diagnostic utility, and potential significance of our approach.
>
> ### Novelty and Positioning
>
> SALVE builds on established components such as sparse autoencoders (SAEs), activation maximization, and weight-space interventions, but its novelty lies in integrating these elements into a unified, post-hoc pipeline for interpretable, permanent model control. This enables capabilities not achievable in isolation:
>
> - Feature discovery → validation → control occurs in a single latent space, grounding edits in semantically validated features.
> - Diagnostics: SALVE’s edits are permanent and sample-dependent, interacting multiplicatively with learned weights. This enables fine-grained control and diagnostics via $\alpha_{\text{crit}}$, supporting per-sample robustness auditing, a capability not natively supported by alternative steering approaches.
>
> To our knowledge, SALVE is the first framework to connect these steps into a methodology for validated, feature-driven weight edits.
>
> ---
>
> ### Q1: Alternative methods
>
> We have expanded Table 1 and the “Related Work” section to include recent SAE-steering approaches and other alternative methods. While an extensive benchmarking is beyond this submission’s scope, we have clarified how SALVE differs from these approaches.
>
> ### Q2: Edit location
>
> We apply edits at the final layer because this provides a clear mapping from SAE-decoded features to classifier decisions. Unlike logit steering, which adjusts logits directly, SALVE modulates feature-to-class connections, enabling sample-dependent behavior. Deeper-layer editing is promising but nontrivial. Such layers correspond to convolutional filters (ResNet) or attention/MLP blocks (ViT), where mapping SAE features to those parameters is less straightforward. We view this as a promising extension and have noted it as future work in the Limitations section.
>
> ### Q3: Benchmarking
>
> We add new experiments directly comparing SALVE to activation steering (revised Appendix A.9). These results show comparable class-suppression performance, while SALVE offers additional capabilities.
>
> **Weight edits vs steering**
> Hook-based SAE steering and other activation-time interventions are runtime-dependent, requiring auxiliary modules during inference. In contrast, SALVE modifies weights directly, producing a single, auditable artifact. This distinction is particularly relevant for compliance-sensitive or deployment-constrained settings.
>
> **Comparison to SAE Steering:**
> Many steering methods, including those used in recent VLM and LLM interventions, apply fixed additive offsets, making them sample-independent in practice. SALVE differs by applying a fixed modulation to weights, but retaining sample-dependency through the activations $x$.
>
> Eq. (1) applies a multiplicative mask: $\tilde W_{ij} = W_{ij} \cdot \text{max}(0, 1 \pm \alpha c_j)$ along a feature direction. This is not equivalent to adding a decoder column to activations. With a fixed $\alpha$ per edit, the net effect remains input dependent because logits combine $\tilde W_{ij}$ with per-sample activations $x_j$. This enables per-sample diagnostics such as $\alpha_{\text{crit}}$, representing a capability not natively supported by standard steering approaches.
>
> These key distinctions are clarified in the revised “Related Work” and “Methods” sections.
>
>
> ### Q4: Evaluation Scope
>
> To assess generalization we have added experiments on CIFAR-100 (revised Appendix A.10). While feature disentanglement is more challenging with greater class diversity, SALVE still discovers meaningful directions and supports effective edits.
>
> Scaling to larger datasets or more complex architectures will likely benefit from more advanced SAE variants (e.g., JumpReLU, gated, top-k). We note that SALVE is SAE-agnostic, and these variants can be integrated without modifying the control or per-sample diagnostic components.
> We expanded the revised Limitations section to clarify this.
>
> ### Q5: ROME
>
> Our ROME adaptation targets the classifier head for consistency with SALVE’s edit location. We agree that mid-layer ROME or MEMIT comparisons would complement future work on deeper-layer edits.
>
> ### Q6: Concept quality
>
> Current validation focuses on model control. Extending the scope to quantitative semantic metrics, such as Network Dissection overlap, TCAV sensitivity, or segmentation IoU, is a promising direction, particularly for studying how SAE architecture and training impact concept quality.
>
> ---
>
> ### Closing
>
> While evaluation is limited to relatively simple models and datasets, we argue this work provides a foundation for future extensions and benchmarking across architectures and modalities. This is beyond the scope of a single paper but represents a promising direction we believe to be of interest to the broader research community.

---

### Official Review · Reviewer_V117 · 2025-11-02

**Soundness:** 2
**Presentation:** 2
**Contribution:** 2
**Rating:** 2
**Confidence:** 3

**Summary:**

This paper introduces SALVE, a framework for discovering, validating, and controlling interpretable features in neural networks through SAEs. The authors train ℓ1-regularized autoencoders on internal activations of vision models to extract sparse feature representations. They validate these features using activation maximization and Grad-FAM visualization technique. The core contribution is a method for performing permanent weight-space edits guided by the learned feature directions, enabling continuous modulation of both class-specific and cross-class features. Experiments demonstrate that the approach enables precise, targeted control over model predictions.

**Strengths:**

- The paper presents a methodologically sound framework with experimental validation across different model architectures.
- The proposed method effectively enables permanent weight-space interventions by directly modifying model weights guided by discovered latent features.

**Weaknesses:**

- While the paper proposes a generic framework integrating sparse autoencoders with weight-space interventions, the individual technical components are largely adaptations of existing methods.
- The comparison with activation steering methods is mentioned but not thoroughly explored empirically.

Please see questions below for details

**Questions:**

- The core techniques—sparse autoencoders for feature extraction, weight-space interventions with minimal edits, and visualization methods—have been explored independently in prior work. The novelty lies primarily in their integration rather than in fundamental technical innovation. I personally think that the paper would benefit from more explicit discussion of how the combined framework differs from and improves upon existing approaches beyond simply integrating known techniques, particularly in comparison to prior work on model editing (e.g., MEMIT) and concept-based interventions that have explored similar ideas of using learned representations to guide weight modifications.
- The paper claims advantages such as "continuous modulation", and "nuanced cross-class edits," but lacks rigorous and comprehensive experimental evidence demonstrating concrete scenarios where permanent weight edits provide clear practical benefits over the more commonly used inference-time steering paradigm. A more systematic comparison showing specific use cases where permanent edits excel—such as computational costs during deployment, capabilities that steering cannot achieve, or finer-grained control compared to recent SAE-based steering approaches—would strengthen the claims about the method's distinct value proposition.

---

> ### Author Response · Authors · 2025-11-19
>
> ### Q1: Novelty and Positioning
>
> SALVE builds on established components such as sparse autoencoders (SAEs), activation maximization, and weight-space interventions, but its novelty lies in integrating these elements into a unified, post-hoc pipeline for interpretable, permanent model control. This enables capabilities not achievable in isolation:
>
> - Feature discovery → validation → control occurs in a single latent space, grounding edits in semantically validated features.
> - Diagnostics: SALVE’s edits are permanent and sample-dependent, interacting multiplicatively with the model’s learned weights. This enables fine-grained control and diagnostics via $\alpha_{\text{crit}}$, supporting per-sample robustness auditing, a capability not natively supported by alternative steering approaches.
>
> To our knowledge, SALVE is the first framework to connect these steps into a methodology for validated, feature-driven weight edits.
>
>
> **Comparison with MEMIT**
>
> MEMIT performs rank-one updates keyed to specific factual associations. SALVE instead modulates general semantic features discovered in an unsupervised manner, enabling continuous and cross-class edits. SALVE further contributes feature-level validation (Grad-FAM) and per-sample sensitivity analysis via $\alpha_{\text{crit}}$, both absent in MEMIT.
>
> **Comparison with Concept-based interventions (INLP, CBMs)**
>
> INLP and CBMs require human-defined concepts or architectural modifications (e.g., concept heads, bottlenecks). SALVE is entirely post-hoc, does not require labels for concept discovery, and provides continuous modulation.
> In the revised “Related Work” section, we make the distinctions between SALVE and these alternative methods more explicit.
>
> ---
>
> ### Q2: Permanent Edits vs Steering
>
> We have added new experiments directly comparing SALVE to activation steering using the same SAE features. These results (included in revised Appendix A.9) show comparable class-suppression performance, while SALVE offers additional capabilities.
>
> **Why “permanent” weight edits over steering:**
> Hook-based SAE steering and other activation-time interventions are runtime-dependent, requiring auxiliary modules during inference. In contrast, SALVE modifies weights directly, producing a single, auditable artifact. This distinction is particularly relevant for compliance-sensitive or deployment-constrained settings.
>
> **Comparison to SAE Steering:**
> Many SAE steering methods, including those used in recent VLM and LLM interventions, apply fixed additive offsets, making them sample-independent in practice. SALVE differs by applying a fixed modulation to weights, but retaining sample-dependency through the activations $x$.
>
> Eq. (1) applies a multiplicative mask: $\tilde W_{ij}=W_{ij} \cdot \text{max}(0, 1 \pm \alpha c_j)$ along a feature direction. This is not equivalent to adding a decoder column to activations. Importantly, with a fixed $\alpha$ per edit, the net effect remains input dependent because logits combine $\tilde W_{ij}$ with per-sample activations $x_j$ (i.e., the edit interacts with each input’s feature composition). This enables per-sample diagnostics such as $\alpha_{\text{crit}}$, representing a capability not natively supported by standard steering approaches.
> These key distinctions are clarified in the revised “Related Work” and “Methods” sections.
>
> ---
>
> ### Evaluation Scope:
>
> To assess generalization beyond small-scale datasets, we have added new experiments on CIFAR-100 (included in revised Appendix A.10). This dataset has substantially greater class diversity than Imagenette. While feature disentanglement is more challenging in this setting, SALVE still discovers meaningful directions and supports effective edits.
>
> Scaling to larger datasets or more complex architectures will likely benefit from SAE variants more widely used in large-scale LLM interpretability (e.g., JumpReLU, gated, top-k). We note that SALVE is SAE-agnostic, and these variants can be integrated without modifying the control or per-sample diagnostic components.
> We have expanded the Limitations and Future Work section in the revision to clarify this.
>
>
> ### Closing:
>
> Our current scope isolates SALVE’s core mechanisms in a controlled setting. We acknowledge our evaluation is limited to relatively simple models and datasets, but argue this work provides a foundation for future research on scaling, benchmarking, and extensions across diverse architectures and modalities. This is beyond the scope of a single paper but represents a promising direction we believe to be of interest to the broader research community.
>
> Our main takeaway is that SALVE, even in its current form, achieves comparable performance to established steering and editing methods while introducing additional diagnostic capabilities.
> We hope these clarifications and the additional experiments address the reviewer’s concerns regarding the distinct advantages of permanent edits and the broader value proposition of our method.

---

### Author Response · Authors · 2025-11-26
**Revised manuscript**

We sincerely thank all reviewers for their thoughtful and constructive feedback, which has improved the clarity, positioning, and empirical scope of the paper. In the revised manuscript we have made several key updates addressing both shared and reviewer-specific concerns:

- **Clearer positioning and novelty.**
  The Related Work section has been expanded to more explicitly distinguish SALVE from activation steering, concept-based interventions, causal editing, and weight-editing approaches such as ROME and MEMIT, clarifying the value proposition of our feature-grounded, post-hoc weight-editing pipeline.

- **Direct benchmarking against activation steering.**
  In response to requests for additional empirical comparison, we added new experiments using the same SAE-derived feature basis, enabling a direct comparison between SALVE and activation steering (Appendix A.9). These results show comparable suppression behavior while highlighting SALVE’s additional diagnostic capabilities.

- **New evaluation on CIFAR-100.**
  To test generalization to more complex settings, we added experiments on CIFAR-100 (Appendix A.10), again comparing SALVE and activation steering using the same latent basis. This provides a clearer picture of edit robustness in higher-diversity datasets.

- **Clarified edit mechanics and sample dependence.**
  The Methods and Related Work sections now more explicitly describe how SALVE’s multiplicative weight modulation differs from activation-time interventions and why it remains input-dependent even with a fixed edit strength.

- **Expanded Limitations and Future Work.**
  We now discuss in greater depth the role of the SAE architecture, the impact of dataset complexity on feature disentanglement, and how stronger SAE variants can be integrated directly into the SALVE pipeline without modifying its control or diagnostic components.

We believe these revisions substantially strengthen the manuscript and directly address the primary concerns raised during review. We kindly invite reviewers to revisit their assessments considering these updates and adjust their evaluations where appropriate.

---

### Meta-Review · Area_Chair_ayjP · 2025-12-11

**Summary:**

The paper introduces SALVE: a method for for discovering, validating, and controlling interpretable features in neural networks through sparse autoencoders.

Major concerns of the reviewers were:
1. Limited technical novelty
2. Insufficient benchmarking
3. Limited evaluation scope
4. Unclear value of the proposed method

All reviewers unanimously scored the paper with 2 (reject), most with confidence 4.

**Reviewer Concerns:**

The authors provided two more appendices, that evaluate their method on CIFAR 100 and added a comparison to activation steering.
They also clarified points about novelty, clarified distinctions from existing methods, and discussed the added value.

However, several key points still remain open:
- The evaluation is still limited to relatively simple models/datasets
- It is unclear whether the method scales well to larger datasets.
- Authors argued that comprehensive benchmarking is out of scope.

**Reviewer Scores:**

Since all reviewers scored with 2 (most with confidence 4), it is very unlikely that all of them would raise the score by 4, in particular given still outstanding issues.

---

### Decision · Program_Chairs · 2026-01-26

Reject